# STYLOS: MULTI-VIEW 3D STYLIZATION WITH SINGLE-FORWARD GAUSSIAN SPLATTING

**Hanzhou Liu**[*]      **Jia Huang**      **Mi Lu**      **Srikanth Saripalli**      **Peng Jiang**[*]
Texas A&M University

## ABSTRACT

We present *Stylos*, a single-forward 3D Gaussian framework for 3D style transfer that operates on unposed content, from a single image to a multi-view collection, conditioned on a separate reference style image. Stylos synthesizes a stylized 3D Gaussian scene without per-scene optimization or precomputed poses, achieving geometry-aware, view-consistent stylization that generalizes to unseen categories, scenes, and styles. At its core, Stylos adopts a Transformer backbone with two pathways: geometry predictions retain self-attention to preserve geometric fidelity, while style is injected via cross-attention to enforce visual consistency across views. With the addition of a voxel-based 3D style loss that aligns aggregated scene features to style statistics, Stylos enforces view-consistent stylization while maintaining geometric coherence. Experiments across multiple datasets demonstrate that Stylos delivers high-quality zero-shot stylization, highlighting the effectiveness of the proposed style-content fusion block, the voxel-level style loss, and the scalability of our framework from single view to large-scale multi-view settings. Our codes are available at https://github.com/HanzhouLiu/Stylos.

## 1 INTRODUCTION

Image guided 3D stylization aims to preserve scene geometry and cross-view consistency while transferring the reference style. With the rise of immersive content, augmented and virtual reality, demand for this capability is growing. Nevertheless, achieving reliable 3D stylization remains challenging and continues to attract significant research attention (Chen et al., 2025).

Early attempts typically focus on 3D representations such as meshes, volumetric data, and point clouds (Kato et al., 2018; Liu et al., 2018; Kim et al., 2019; 2020; Guo et al., 2021; Cao et al., 2020). The introduction of implicit neural radiance fields (NeRF) (Mildenhall et al., 2020), enables higher-fidelity rendering, and subsequent works extended it to artistic stylization (Zhang et al., 2022; Huang et al., 2022; Nguyen-Phuoc et al., 2022; Bao et al., 2022; Liu et al., 2023), but typically require costly per-scene optimization, constraining their generalization to unseen scenes. More recently, 3D Gaussian Splatting (3DGS) has emerged as a promising explicit representation that combines high reconstruction quality with real-time rendering efficiency (Kerbl et al., 2023). Stylization approaches built on 3DGS achieve efficient multi-view consistency (Liu et al., 2024; Mei et al., 2024; Galerne et al., 2025), yet still struggle to generalize beyond scene-specific training.

In contrast, we introduce *Stylos* (meaning *pens* in French), a single-forward framework for 3D style transfer that eliminates the need for per-scene optimization and precomputed camera parameters, and generalizes effectively to novel categories, scenes, and styles. Stylos employs a shared Transformer backbone with two pathways: content and style images are projected into a shared feature space, where content retains self-attention for geometric reasoning, and style is injected via the Cross-Block modules that are primarily composed of a self-attention layer followed by cross-attention. Geometry-related attributes, such as depth, camera intrinsics, and extrinsics, are derived from backbone features, whereas style conditioning guides the prediction of color coefficients. These outputs are estimated through prediction heads that serve as the interface between feature space and the final Gaussian representation. A representative style loss function in 2D style transfer is based on feature distribution alignment (Li et al., 2017; Huang & Belongie, 2017; Jing et al., 2019; Singh et al.,

---

* Equal Contribution

2021), which operate on image-level statistics and do not explicitly enforce multi-view or structural consistency required for 3D stylization. To address this, we explore alternative objectives and propose a voxel-level 3D style loss that aligns aggregated scene features with style statistics, providing stronger view-consistent stylization while preserving geometric fidelity.

We evaluate Stylos across different scenarios, including category-level transfer and cross-scene generalization. Our assessment spans challenging real-world benchmarks, where Stylos produces stylized renderings with high visual fidelity and consistent geometry, demonstrating robustness even in previously unseen environments. Our main contributions are threefold:

- We propose a shared-backbone design with two pathways: geometry predictions retain self-attention for geometric reasoning, while style is injected through cross-attention.
- We introduce a voxel-level 3D style loss that aligns aggregated scene features with style statistics, enforcing cross-view coherence and geometry-aware stylization.
- We develop Stylos, a single-forward-pass pipeline for 3D style transfer from unposed inputs, scaling from a single to hundreds of views with a single style image, and achieving zero-shot generalization to unseen categories, scenes, and styles.

## 2 RELATED WORK

### 2.1 POSE-FREE 3D RECONSTRUCTION

3D reconstruction from unposed multi-view images or videos has drawn significant attention with the development of feed-forward models, including NeRF-based methods (Smith et al., 2023; Hong et al., 2024) and 3DGS-based approaches (Li et al., 2024; Kim, 2025). DUSt3R (Wang et al., 2024), a Transformer-based framework, advances this direction by enabling pointmap estimation directly from image pairs. Its extensions (Wang & Agapito, 2024; Yang et al., 2025; Wang et al., 2025c; Cabon et al., 2025) further reduce reliance on global alignment procedures and support a varying number of input views. VGGT (Wang et al., 2025a), a recent 3D foundation model, revolutionizes this line of work by jointly predicting camera parameters, depth, point maps, and tracks from one to hundreds of views in a single forward pass, without any pose optimization. As a follow-up to VGGT, AnySplat (Jiang et al., 2025) introduces a rendering head and complements this design with a novel feed-forward Gaussian splatting pipeline, enabling the prediction of Gaussian primitives along with depth and camera parameters from uncalibrated images.

### 2.2 3D STYLE TRANSFER AND LOSSES

3D stylization methods can be classified based on geometric representations, such as mesh (Kato et al., 2018; Liu et al., 2018), volumetric data (Kim et al., 2019; 2020; Guo et al., 2021), point coulds (Cao et al., 2020; Huang et al., 2021). In addition, neural radiance fields (NeRF) (Mildenhall et al., 2020), using an MLP to implicitly learn a static 3D scene, has inspired a range of stylization methods based on stylized view supervision or auxiliary networks (Nguyen-Phuoc et al., 2022; Zhang et al., 2022; Chiang et al., 2022; Huang et al., 2022). StylizedNeRF (Huang et al., 2022) addresses the domain gap between style images and NeRF by introducing a mutual learning framework. StyleRF (Liu et al., 2023) further enables zero-shot transfer by applying style transformations in radiance-field feature space. However, training NeRFs for scene reconstruction and stylization is relatively computationally expensive, making NeRF-based 3D stylization far from real-time.

More recently, 3D Gaussian Splatting (3DGS) (Kerbl et al., 2023) has emerged as an efficient alternative to NeRF for 3D scene modeling. Subsequent works refine 3DGS for efficient stylization (Mei et al., 2024; Kovács et al., 2024; Yu et al., 2024; Zhang et al., 2025; Galerne et al., 2025; Lin et al., 2025). StyleGaussian (Liu et al., 2024) extends 3DGS with efficient feature rendering to enable real-time zero-shot stylization. However, despite being significantly faster to optimize than NeRF, 3DGS-based approaches still require per-scene fitting, which poses a fundamental barrier to achieving truly real-time 3D stylization. Styl3R (Wang et al., 2025b), the closest contemporaneous related work to ours, introduces a feedforward framework for instant 3D stylized reconstruction. While effective and efficient, Styl3R is primarily designed for 2–8 input views and does not specifically target strong multi-view consistency among the rendered stylization results.

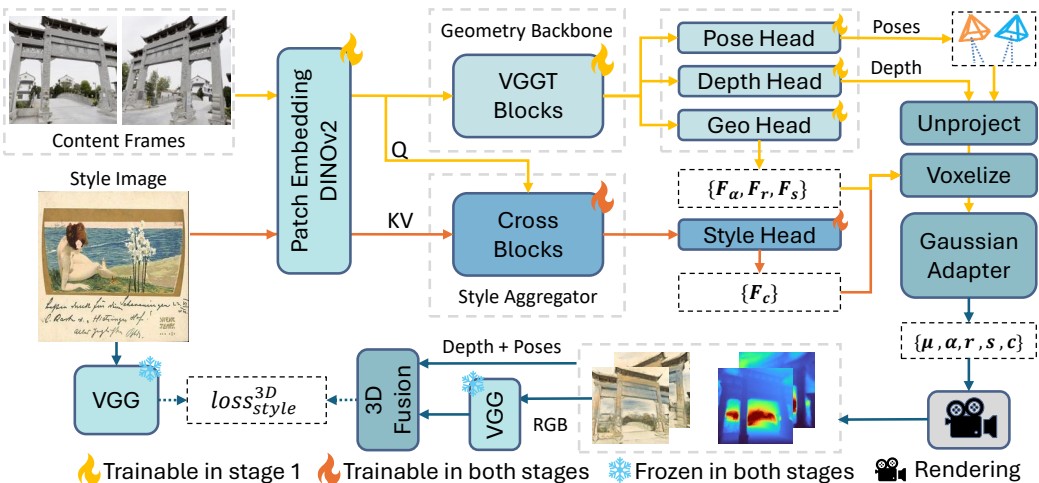

Figure 1: **Architecture overview**. Given multi-view content inputs and a style reference, Stylos enables instant 3D stylization without scene-specific training or post-optimization. Its core Cross-Block module facilitates style injection by integrating a cross-attention layer between self-attention and MLP. The proposed 3D style loss matches voxelized 3D features with 2D style statistics.

Underlying these architectures, style transfer objectives play a crucial role. Classical style transfer relies on Gram-matrix correlations (Gatys et al., 2016), while channel-wise statistics such as mean and variance (Li et al., 2017) offer efficient alternatives, forming the basis of AdaIN (Huang & Belongie, 2017). Extensions to video and multi-view stylization (Ruder et al., 2016; Gupta et al., 2017; Nguyen-Phuoc et al., 2022; Zhang et al., 2022) address temporal and cross-view consistency, and CLIP-based losses (Radford et al., 2021) introduce semantic alignment with text or image prompts. However, most objectives remain inherently 2D and cannot guarantee consistency across complex 3D scenes. To this end, we extend statistics-based style losses into voxel space, introducing a 3D-aware objective that aligns feature distributions after multi-view fusion.

## 3 METHOD

We propose *Stylos*, a transformer-based framework for stylized 3D scene reconstruction. Given a style reference image and one or more content views (also referred to as context views), Stylos predicts a set of stylized 3D Gaussian primitives together with camera parameters, enabling faithful reconstruction of the observed scene while transferring the desired style. We first formulate the problem in Sec. 3.1, then present the network architecture in Sec. 3.2, and finally detail our training strategy in Sec. 3.3 and the used style loss in Sec. 3.4.

### 3.1 PROBLEM FORMULATION

We aim to disentangle geometry and style within a unified 3D scene representation. The input to Stylos consists of a set of $N$ context views $\{I_i\}_{i=1}^N$ of a scene, with $N \geq 1$ and $I_i \in \mathbb{R}^{H \times W \times 3}$, together with a single style reference image $S \in \mathbb{R}^{H \times W \times 3}$. The context views provide geometric cues, while the style image specifies the desired aesthetic.

Formally, Stylos defines a conditional mapping,

$$f_\theta : (\{I_i\}_{i=1}^N, S) \ \mapsto \ (G, \{g_i\}_{i=1}^N),$$

where the scene representation $G = \{(p_m, c_m)\}_{m=1}^M$ is parameterized by $M$ anisotropic 3D Gaussians. Each Gaussian has geometry attributes $p_m = (\mu_m, \alpha_m, r_m, s_m)$, comprising 3D position $\mu_m \in \mathbb{R}^3$, opacity $\alpha \in \mathbb{R}^+$, orientation quaternion $r_m \in \mathbb{R}^4$, and anisotropic scale $s_m \in \mathbb{R}^3$, as well as a style-dependent color embedding $c_m \in \mathbb{R}^{3 \times (k+1)^2}$ represented with spherical harmonics of degree $k$. In addition, Stylos jointly predicts camera parameters $\{g_i \in \mathbb{R}^9\}_{i=1}^N$ for each input view, leveraging pose cues estimated by the VGGT backbone.

## 3.2 Network Architecture

Stylos employs a geometry backbone that alternates frame-wise and global attention to process multiple input views and infer geometry-related parameters such as positions, scales, orientations, and opacities. To enable stylization, we introduce a dedicated conditioning branch, where the *Style Aggregator* fuses content and style features through cross-attention and predicts style-aware color embeddings for each Gaussian. In this way, geometry remains derived solely from the backbone, while style representation is conditioned on the style reference. We next detail the geometric backbone in Sec. 3.2.1, the style aggregation module in Sec. 3.2.2, and the prediction heads in Sec. 3.2.3.

### 3.2.1 Geometric Backbone

The geometric backbone follows the alternating-attention design of VGGT (Wang et al., 2025a), which interleaves frame and global self-attention layers to capture both intra-frame structure and cross-view consistency. This component is kept unchanged to retain strong geometric reasoning, serving as the foundation on which we integrate style information.

### 3.2.2 Style Aggregator

The Style Aggregator builds on the geometric backbone inherited from VGGT, which is composed of standard Transformer Block containing self-attention followed by feed-forward layers. To enable style conditioning, we replace this standard block with an adapted *cross fusion block* (CrossBlock), which inserts a cross-attention operation between the self-attention and MLP stages (Deng et al., 2022). In CrossBlock, content tokens serve as queries ($\mathcal{Q}$) while style tokens provide keys and values ($\mathcal{KV}$). This structure allows the content representations to be refined by their own spatial context (via the preserved self-attention), while being explicitly conditioned on the target style (via the inserted cross-attention) before the final feed-forward projection.

**Cross Fusion Blocks.** Let $\mathcal{Q}_{b,v} \in \mathbb{R}^{L_q \times C}$ denote the content tokens extracted from the $v$-th view of the $b$-th sample, and let $\mathcal{KV}_b \in \mathbb{R}^{L_{kv} \times C}$ denote the style tokens. We use $\mathrm{CrossBlock}(\mathcal{Q}, \mathcal{KV})$ to denote our block operator that updates $\mathcal{Q}$ using $\mathcal{KV}$ as key–value pairs. Depending on how the query set is formed, we consider two primary topological strategies.

**(1) Frame CrossBlock.** Each view independently interacts with the style tokens,

$$\widetilde{\mathcal{Q}}_{b,v} = \mathrm{CrossBlock}(\mathcal{Q}_{b,v}, \mathcal{KV}_b). \tag{1}$$

Since the internal self-attention of the block operates only on $\mathcal{Q}_{b,v}$, this strategy preserves view-specific geometric structure but prevents information propagation between different views.

**(2) Global CrossBlock.** We first concatenate all views to obtain a global sequence $\mathcal{Q}_b^{\mathrm{global}} \in \mathbb{R}^{VL_q \times C}$ by $\mathcal{Q}_b^{\mathrm{global}} = \mathrm{Concat}_{v=1}^{V} \mathcal{Q}_{b,v}$, and perform the block operation simultaneously,

$$\widetilde{\mathcal{Q}}_b^{\mathrm{global}} = \mathrm{CrossBlock}(\mathcal{Q}_b^{\mathrm{global}}, \mathcal{KV}_b). \tag{2}$$

We then reshape the updated tokens back to per-view tensors, $\widetilde{\mathcal{Q}}_{b,v}$, by splitting $\widetilde{\mathcal{Q}}_b^{\mathrm{global}}$ along the sequence dimension. This approach enables long-range reasoning: the internal self-attention ensures multi-view geometric consistency, while the cross-attention broadcasts style information globally.

**(3) Hybrid CrossBlocks.** We first apply Frame CrossBlock to refine each view independently as in Eq. 1, and then perform Global CrossBlock by concatenating the resulting $\widehat{\mathcal{Q}}_{b,v}$ across views and applying the same operator as in Eq. 2. This hybrid design combines per-view refinement with multi-view aggregation. We direct readers to the released codes for implementations.

### 3.2.3 Prediction Heads

To connect the geometric backbone and Style Aggregator with the Gaussian scene, we introduce prediction heads that translate feature tokens into explicit parameters. These heads serve as modular interfaces, keeping geometry and style distinct in feature space while ensuring their integration in the final Gaussian representation. We next describe the individual heads in detail.

**Geometry Head.** Geometric backbone features are passed through a DPT-style regression head that outputs the Gaussian geometry parameters $p_m = (\mu_m, s_m, r_m, \alpha_m)$, i.e., position, scale, orientation,

and opacity (as defined in Sec. 3.1). By relying on this established design, structural predictions are derived from backbone features alone, without direct influence from style conditioning.

**Style Head.** The outputs of the Style Aggregator are subsequently processed by a color head to predict the spherical-harmonic coefficients $c_m$ that define appearance. This pathway injects style information directly into Gaussian colors while leaving the geometry parameters $p_m$ unaffected, enabling the two factors to be recombined seamlessly at the Gaussian level.

**Auxiliary Heads, Adapter, and Voxelization.** We employ the existing VGGT camera head to estimate camera intrinsics and extrinsics, and a depth head to predict scene geometry cues, which are unprojected into 3D anchors for Gaussian placement (Kerbl et al., 2023; Ren et al., 2025). A Gaussian adapter then consolidates geometry and style outputs into a unified set of primitives $\{(p_m, c_m)\}_{m=1}^M$ for differentiable rendering. Finally, to reduce redundancy and balance density, we follow the voxelization step introduced by AnySplat (Jiang et al., 2025), nearby points are clustered within a discretized 3D grid and fused using confidence-aware weighting. This operation depends only on the unprojected 3D points and features, and is independent of camera parameters.

## 3.3 TRAINING STRATEGY

We adopt a two-stage training strategy for structure-aware stylization.

**Stage 1: Geometry Pretraining.** We initialize the geometric backbone of Stylos with VGGT weights (Wang et al., 2025a) and train the network end-to-end to learn geometry and photometric appearance. To avoid trivial identity mapping and improve robustness to color variations, one input view is randomly selected and color-jittered as the style reference. A frozen VGGT teacher provides pose and depth supervision. The objective combines reconstruction and distillation,

$$\mathcal{L}_{\text{stage1}} = \mathcal{L}_{\text{rec}} + \lambda_{\text{distill}}\,\mathcal{L}_{\text{distill}}.$$

**Stage 2: Stylization Fine-tuning.** We freeze all geometry-related modules and only update the Style Aggregator and the color head. Following ArtFlow (An et al., 2021), we use feature-level style and content losses in VGG space, matching channel-wise statistics for style and feature activations for content. We further extend these objectives with a 3D voxel-space style loss for cross-view consistency, and add a CLIP-based loss for semantic alignment. A total variation (TV) regularizer is also included to suppress high-frequency artifacts and stabilize optimization. The total loss is,

$$\mathcal{L}_{\text{stage2}} = \mathcal{L}_{\text{rec}} + \lambda_{\text{style}}\,\mathcal{L}_{\text{style}}^{3D} + \lambda_{\text{cnt}}\,\mathcal{L}_{\text{content}} + \lambda_{\text{clip}}\,\mathcal{L}_{\text{clip}} + \lambda_{\text{tv}}\,\mathcal{L}_{\text{TV}}.$$

In our experiments, we set $\lambda_{\text{distill}} = 1.0$, and use $\{\lambda_{\text{style}}, \lambda_{\text{cnt}}, \lambda_{\text{clip}}, \lambda_{\text{tv}}\} = \{1.0, 0.1, 1.0, 10.0\}$

## 3.4 STYLE LOSSES

We denote by $\mathcal{R}_{b,v}^l \in \mathbb{R}^{C_l \times H_l \times W_l}$ the VGG feature map of the rendered image for the $b$-th scene and $v$-th view at layer $l$, and by $\mathcal{S}_b^l \in \mathbb{R}^{C_l \times H_l \times W_l}$ the feature map of the single style image. Building on the Batch Normalization statistics method (Li et al., 2017), which interprets BN mean and variance as style descriptors, we progressively extend it matching beyond standard 2D settings. This progression starts from independent image-level matching, moves to multi-view feature aggregation that promotes global style coherence across views, and culminates in a voxel-space loss that directly constrains the fused 3D representation. Next, we formulate the three losses studied in this paper.

**Image-Level Style Loss.** The simplest baseline aligns each rendered frame independently with the style reference, which encourages per-frame stylization but does not enforce multi-view consistency,

$$\mathcal{L}_{\text{sty}}^{\text{img}} = \frac{1}{BV} \sum_{b=1}^B \sum_{v=1}^V \sum_{l=1}^5 \alpha_l \Big( \|\mu(\mathcal{R}_{b,v}^l) - \mu(\mathcal{S}_b^l)\|_2^2 + \|\sigma(\mathcal{R}_{b,s}^l) - \sigma(\mathcal{S}_b^l)\|_2^2 \Big), \qquad (3)$$

where $\alpha_l$ denotes the weight at each level $l$, $\mu$ is the mean and $\sigma$ is the standard deviation.

**Scene-Level Style Loss.** To introduce consistency into multi-view 3D stylization, we concatenate the per-view features $\{\mathcal{R}_{b,v}^l\}_{v=1}^V$ along the spatial dimension to form $\widetilde{\mathcal{R}}_b^l$, and calculate statistics on this aggregated map. However, this method still operates in 2D feature space and is computed by,

$$\mathcal{L}_{\text{sty}}^{\text{scn}} = \frac{1}{B} \sum_{b=1}^B \sum_{l=1}^5 \alpha_l \Big( \|\mu(\widetilde{\mathcal{R}}_b^l) - \mu(\mathcal{S}_b^l)\|_2^2 + \|\sigma(\widetilde{\mathcal{R}}_b^l) - \sigma(\mathcal{S}_b^l)\|_2^2 \Big). \qquad (4)$$

---

**Algorithm 1** 3D Voxel-based AdaIN Style Loss

---

**Require:** Rendered features $\mathcal{R}$, Style features $\mathcal{S}$, 3D points $\mathcal{P}$, Confidence $\mathcal{C}$, Valid mask $\mathcal{M}$
**Ensure:** 3D style loss $\mathcal{L}_{sty}^{\text{3D}}$
 1: Initialize $\mathcal{L}_{sty}^{3D} \leftarrow 0$
 2: **for** $b = 1$ to $B$ **do**
 3:     **for** $l = 1$ to $5$ **do**
 4:         Resize $(\mathcal{C}, \mathcal{M}, \mathcal{P})$ to match the resolution of $\mathcal{R}_b^l$
 5:         *// Next, fuse all views into a single voxel grid*
 6:         $\mathcal{G}_b^l \leftarrow \text{VoxelizeAndFuse}\left(\{\mathcal{R}_{b,v}^l\}_{v=1}^V, \{\mathcal{P}_{b,v}\}_{v=1}^V, \{\mathcal{C}_{b,v}\}_{v=1}^V, \{\mathcal{M}_{b,v}\}_{v=1}^V\right)$
 7:         *// Next, compute the BN statistics (mean and standard deviation) of two feature maps*
 8:         $(\mu_g, \sigma_g) \leftarrow \text{MeanStd}(\mathcal{G}_b^l), (\mu_s, \sigma_s) \leftarrow \text{MeanStd}(\mathcal{S}_b^l)$
 9:         $\mathcal{L}_{sty}^{3D} \leftarrow \mathcal{L}_{sty}^{3D} + \alpha_l \|\mu_g - \mu_s\|_2^2 + \alpha_l \|\sigma_g - \sigma_s\|_2^2$
10:     **end for**
11: **end for**
12: **return** $\mathcal{L}_{sty}^{3D} = \lambda_{sty} \mathcal{L}_{sty}^{3D}$

---

**Voxel-level 3D Style Loss.** Finally, to further enforce multi-view consistency, we fuse multi-view features into a voxel grid using differentiable unprojection, where features from different views are accumulated into spatial bins of a discretized 3D volume. Let $\mathcal{G}_b^l$ denote the voxelized features for the $b$-th scene at layer $l$. We then compute style statistics directly in voxel space,

$$\mathcal{L}_{\text{sty}}^{\text{3D}} = \frac{1}{B} \sum_{b=1}^B \sum_{l=1}^5 \alpha_l \left( \|\mu(\mathcal{G}_b^l) - \mu(\mathcal{S}_b^l)\|_2^2 + \|\sigma(\mathcal{G}_b^l) - \sigma(\mathcal{S}_b^l)\|_2^2 \right). \tag{5}$$

By operating on voxelized features, this loss explicitly encodes geometry and enforces style consistency across both views and the underlying 3D structure. We provide pseudo codes in Algorithm 1.

## 4 EXPERIMENT

**Datasets.** We evaluate cross-category generalization on the CO3D (Reizenstein et al., 2021) dataset by training on 17 categories and testing on 3 held-out ones, and cross-scene generalization by training on the full DL3DV-10K (Ling et al., 2024) and testing on Tanks & Temples (Knapitsch et al., 2017). Style images are provided by WikiArt (WikiArt, 2025) and DELAUNAY (Gontier et al., 2023), with 50 diverse style images reserved as unseen styles that are never used during training.

**Baselines.** We compare *Stylos* with recent 3D stylization models. The baselines include, (1) a per-scene training and zero-shot method StyleGaussian (Liu et al., 2024), (2) per-scene and per-style optimization approaches G-Style (Kovács et al., 2024), StylizedGS (Zhang et al., 2025), and SGSST (Galerne et al., 2025), and (3) the closest related work Styl3R (Wang et al., 2025b).

**Evaluation Metrics.** Our evaluation covers three aspects. (1) To assess geometry reconstruction in the style-free setting, we report *PSNR*, *SSIM*, and *LPIPS* (Zhang et al., 2018) between content views and predictions, conditioned on the original first frame of each scene as the style input. (2) To measure stylization consistency, following prior work (Chiang et al., 2022), we compute *LPIPS* and *RMSE* in both short-range and long-range settings. (3) To further evaluate stylization quality, we report *ArtScore* (Chen et al., 2024), a recent metric specifically designed for reference-free evaluation of artness in generated images, and *ArtFID* (Wright & Ommer, 2022; Chung et al., 2024).

### 4.1 ABLATION STUDY

In this section, we analyze the impact of each design choice on the CO3D dataset. To ensure fairness, all variants are trained with identical configurations in each experimental round.

**Discussions about CrossBlock Designs.** Take the pizza scene for example. Table 1 shows that the baseline method, combining Frame and Global CrossBlock, yields a PSNR of 19.78 dB and LPIPS of 0.3326. The frame variant produces slightly lower PSNR and higher LPIPS. By contrast, applying the Global CrossBlock significantly boosts the PSNR value by 0.79 dB and achieves consistently

Table 1: Ablation on the style-content fusion module, comparing Frame, Global and hybrid Cross-Block designs. The first frame of each content scene is used as the pseudo style reference. Reconstruction quality is evaluated with PSNR↑, SSIM↑, and LPIPS↓ on the CO3D dataset.

| Strategies | | Skateboard | | | Pizza | | | Donut | | |
|---|---|---|---|---|---|---|---|---|---|---|
| Global | Frame | PSNR↑ | SSIM↑ | LPIPS↓ | PSNR↑ | SSIM↑ | LPIPS↓ | PSNR↑ | SSIM↑ | LPIPS↓ |
| ✓ | ✓ | 21.12 | 0.6858 | 0.2821 | 19.78 | 0.5939 | 0.3326 | 21.39 | 0.7198 | 0.3264 |
| | ✓ | 20.93 | 0.6917 | 0.2912 | 19.72 | 0.5940 | 0.3405 | 21.40 | 0.7167 | 0.3340 |
| ✓ | | **21.68** | **0.7043** | **0.2684** | **20.57** | **0.6177** | **0.3110** | **22.09** | **0.7362** | **0.3125** |

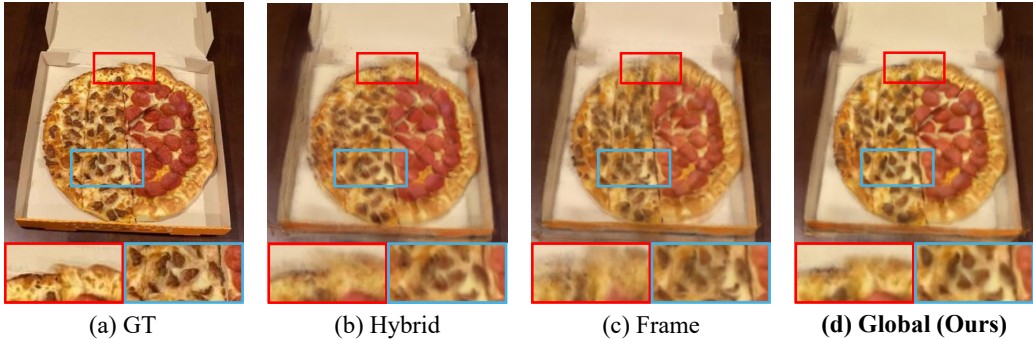

(a) GT      (b) Hybrid      (c) Frame      **(d) Global (Ours)**

Figure 2: CO3D pizza scene comparing different CrossBlock designs.

improved metrics. Qualitative comparisons in Fig. 2 highlight these differences. Frame CrossBlock produces a poorly defined crust with noticeable artifacts, the hybrid variant recovers part of the structure but leaves the box edges blurred, while Global CrossBlock delivers the most faithful result, with clear toppings and a well-preserved crust boundary.

**Image vs. Scene vs. 3D Style Losses.** As shown in Table 2, both scene-level and 3D style losses clearly outperform the image-level baseline in terms of artistic quality. Overall, the 3D losses exhibit superior and more stable consistency performance. These results indicate that enforcing style consistency in a voxel-level yields more coherent stylized textures across views. We observe that the image-level supervision may fail to transfer style. For example, in Fig. 3, the donut surface is not well synthesized with the expected style, whereas the scene and 3D losses successfully apply the target style. The 3D style loss produces sharper boundaries and a stronger sense of 3D geometry, as observed in the skateboard and pizza examples, leading to the most coherent multi-view stylization. We release these model weights to support reproducibility and further analysis.

**From a Single View to Dozens of Views.** While Stylos can process up to dozens (even hundreds) of views, we observe a gradual decrease in visual quality once the number of views per batch (denoted as "# views / batch") exceeds 32. As shown in Fig. 4, very small batches (1 view) fail to synthesize parts of the scene like the bench, while overly large batches (64 views) introduce edge artifacts on the tall building, potentially due to the gap from our training settings (no more than 24 views). Additionally, we provide A.2 Fig. 7 to show the efficiency trend with respect to the number of views.

Table 2: Comparison of consistency and artistic quality among different style loss designs on CO3D. The frame stride is set to 3, and 15 held-out scenes are randomly selected for evaluation.

| Style Loss | Short-range | | Long-range | | ArtScore↑ |
|---|---|---|---|---|---|
| | LPIPS↓ | RMSE↓ | LPIPS↓ | RMSE↓ | |
| Image loss (Eq. 3) | 0.048 | 0.038 | 0.157 | **0.142** | 4.78 |
| Scene loss (Eq. 4) | **0.047** | 0.036 | 0.156 | 0.148 | 9.12 |
| **3D loss (Eq. 5)** | **0.047** | **0.034** | **0.153** | **0.142** | **9.15** |

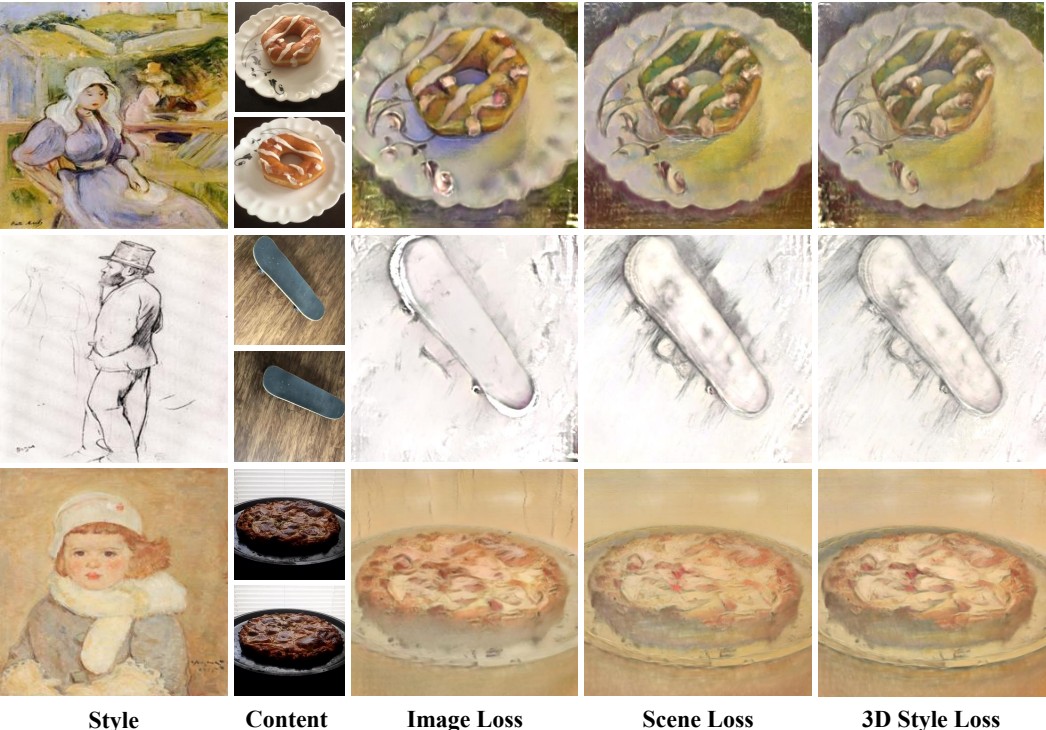

| Style | Content | Image Loss | Scene Loss | 3D Style Loss |

Figure 3: Comparison of style losses on unseen donut, skateboard, and pizza scenes from the CO3D dataset. Both scene and 3D style losses yield cleaner stylized textures compared to image-level matching, while the 3D loss further conveys a stronger sense of 3D geometry. We encourage readers to Appendix Fig. 8-10 for more visual comparisons under varing scenes and styles.

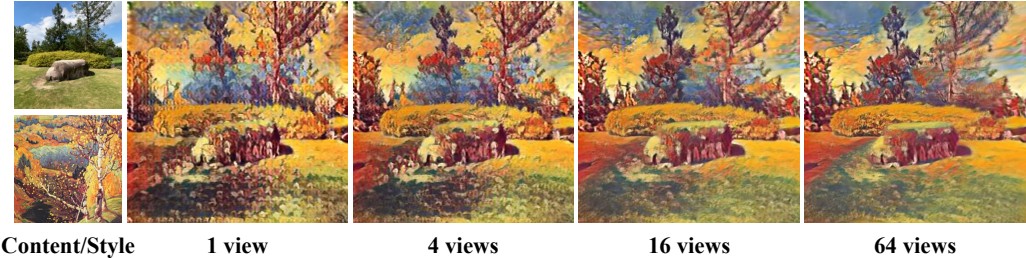

| Content/Style | 1 view | 4 views | 16 views | 64 views |

Figure 4: Effect of varying # views / batch on the Lighthouse scene from Tanks and Temples.

## 4.2 COMPARISON WITH STATE-OF-THE-ART

To retrain StyleGaussian (Liu et al., 2024), G-Style (Kovács et al., 2024), StylizedGS (Zhang et al., 2025), and SGSST (Galerne et al., 2025), we strictly follow their released codes. Since the pretrained-weights of StyleGaussian (Liu et al., 2024) and Styl3R (Wang et al., 2025b) are publicly available, we use them to generate the visual results. Stylos is trained once on DL3DV (Ling et al., 2024), and tested in a zero-shot manner without prior knowledge of either the scenes or styles.

**Quantitative Evaluation.** As shown in Table 3, Stylos achieves strong and stable consistency scores, ranking the first across all consistency metrics and all four scenes. This indicates that Stylos provides markedly improved cross-view stylization consistency. Furthermore, Table 4 shows that Stylos attains either the best or second-best artistic metric values, as rated by ArtScore and ArtFID, on all four Tanks and Temples scenes (Knapitsch et al., 2017), while maintaining the fastest stylization speed. Overall, Stylos demonstrates a favorable balance between visual quality, consistency, and efficiency, suggesting its potential for practical real-time 3D stylization.

Table 3: For consistency comparisons, we report short-range and long-range LPIPS↓ and RMSE↓ on the four scenes from the Tanks & Temples dataset. We clarify experiment details in A.4. In the following tables, the best results are **highlighted** and the second best results are underlined. Each stylization method category is visualized with a distinct color. The proposed *Stylos* demonstrates improved short-range and long-range consistency scores across the four scenes.

| Method | Train | | Truck | | M60 | | Garden | |
|---|---|---|---|---|---|---|---|---|
| | LPIPS↓ | RMSE↓ | LPIPS↓ | RMSE↓ | LPIPS↓ | RMSE↓ | LPIPS↓ | RMSE↓ |
| *Short-range consistency* | | | | | | | | |
| StyleGaussian 2024 | 0.033 | 0.038 | 0.031 | 0.034 | 0.038 | 0.037 | 0.069 | 0.061 |
| G-Style 2024 | 0.042 | 0.052 | 0.032 | 0.035 | 0.038 | 0.034 | 0.066 | 0.070 |
| SGSST 2025 | 0.038 | 0.047 | 0.039 | 0.047 | 0.044 | 0.049 | 0.084 | 0.090 |
| Styl3R 2025b | – | – | 0.061 | 0.036 | 0.066 | 0.040 | 0.105 | 0.067 |
| *Stylos* (ours) | **0.030** | **0.026** | **0.028** | **0.021** | **0.035** | **0.024** | **0.047** | **0.044** |
| *Long-range consistency* | | | | | | | | |
| StyleGaussian 2024 | 0.067 | 0.072 | 0.086 | 0.077 | 0.091 | 0.091 | 0.177 | 0.141 |
| G-Style 2024 | 0.098 | 0.120 | 0.095 | 0.093 | 0.104 | 0.095 | 0.180 | 0.175 |
| SGSST 2025 | 0.087 | 0.108 | 0.119 | 0.120 | 0.130 | 0.128 | 0.221 | 0.222 |
| Styl3R 2025b | – | – | 0.116 | 0.100 | 0.147 | 0.143 | 0.146 | 0.145 |
| *Stylos* (ours) | **0.051** | **0.056** | **0.074** | **0.069** | **0.083** | **0.082** | **0.139** | **0.134** |

StylizedGS (Zhang et al., 2025) is not included in quantitative comparisons due to its multiple failure cases observed on our test styles. Nevertheless, its quantitative results are reported in A.4 Table 5 and Table 6 for readers' reference. Reproduced results are available at https://github.com/HanzhouLiu/Stylos.

Table 4: Stylization quality, as measured by ArtScore and ArtFID (abbreviated as Score and FID respectively), and stylization time comparisons with recent 3D stylization models. *Stylos* achieves consistently favorable metric scores across the four scenes. Additionally, we follow StyleID (Chung et al., 2024) and calculate additional metrics as reported in A.4 Table 5 and Table 6.

| Method | Train | | Truck | | M60 | | Garden | | Time↓ |
|---|---|---|---|---|---|---|---|---|---|
| | Score↑ | FID↓ | Score↑ | FID↓ | Score↑ | FID↓ | Score↑ | FID↓ | |
| StyleGaussian 2024 | 0.78 | 52.79 | 5.76 | 44.93 | 8.63 | 47.48 | **9.38** | 41.14 | 165 m[*] |
| G-Style 2024 | **9.52** | **23.24** | 9.67 | **22.15** | **9.73** | **22.36** | 8.98 | **25.76** | 14.7 m[*] |
| SGSST 2025 | 1.84 | 38.24 | 5.34 | 32.34 | 5.26 | 38.73 | 4.89 | 33.54 | 35.2 m[*] |
| Styl3R 2025b | – | – | 2.94 | 34.11 | 2.96 | 29.86 | 4.09 | 38.28 | 0.16 s[†] |
| *Stylos* (ours) | 9.50 | 26.40 | **9.70** | 28.71 | 9.37 | 27.44 | 9.34 | 28.06 | **0.05 s[†]** |

[*] For methods with per-scene fitting required, the training phase accounts towards stylization time (see A.4).
[†] As to single-forward 3D stylization approaches, training is not considered for stylization time.

**Qualitative Evaluation.** Taking the truck scene with the *desert-town* style (yellow foreground objects with blue backgrounds) in Fig. 5 as an example, we observe that nearlly all existing approaches could reasonably produce structured results. However, their differences are discussed as follows. The zero-shot method StyleGaussian fail to generate color-consistent scenes, several portions of the truck retain undesired color blocks. Per-scene and per-style fitting approaches such as G-Style, StylizedGS, and SGSST often fail to achieve complete style transfer, the truck door remaining predominantly blue instead of adopting the target yellow hue. The other feedforward solution Styl3R similarly struggles to propagate the correct yellow color throughout the truck. In contrast, Stylos successfully renders the truck in coherent yellow while preserving clean geometry, and correctly assigns blue tones to the distant background regions, producing an appearance that closely resembles the intended desert-town aesthetic. These observations underscore the strength of Stylos in achieving both faithful style expression and robust 3D structural preservation.

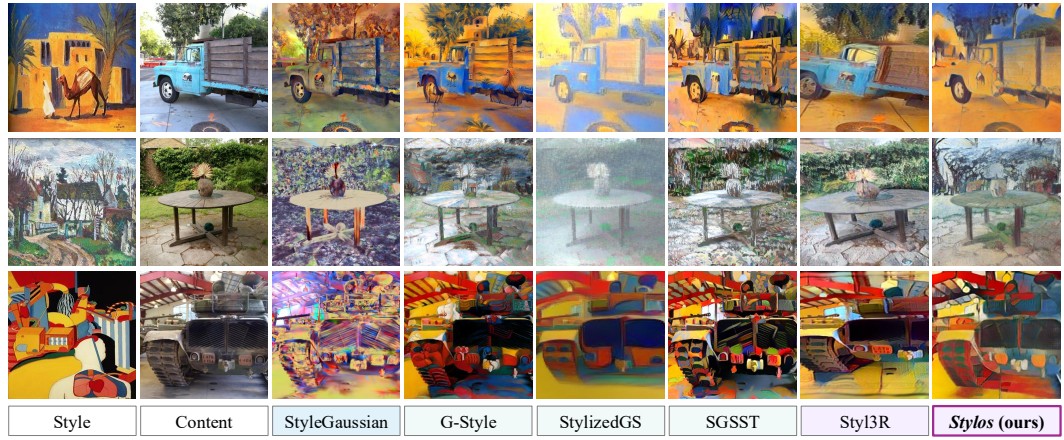

Figure 5: Visual comparisons between *Stylos* and recent 3D stylization approaches. *Stylos* successfully transfers diverse artistic styles to the scenes while preserving fine structural details.

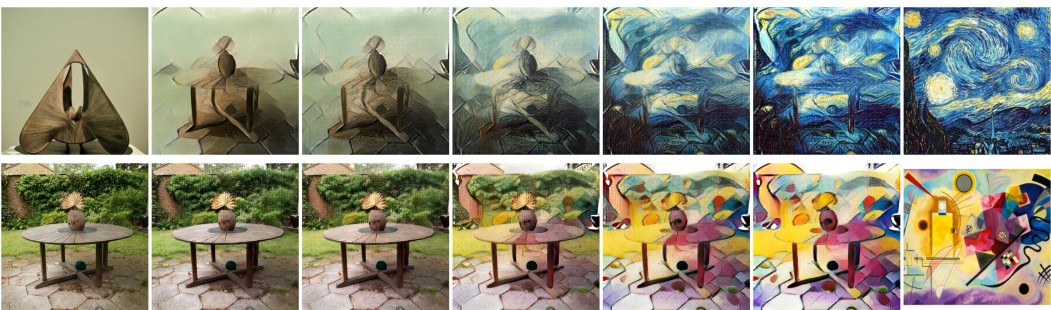

Figure 6: Evaluation of multi-style blending and controllable stylization. Top: interpolation between two style embeddings. Bottom: interpolation between the content embedding and a style embedding.

## 4.3 EVALUATION OF MULTI-STYLE BLENDING AND STYLIZATION CONTROL

We validate the model's controllable stylization capability through two interpolation experiments. First, by linearly interpolating between the embeddings of two distinct style images, we observe smooth, coherent transitions that demonstrate the model's natural support for multi-style blending (see Fig. 6). Second, because the model is trained to reconstruct the original appearance when the style image matches the content image, we interpolate between the reconstruction (content) embedding and a stylized embedding Fig. 6. This produces a continuous spectrum of outputs with gradually increasing stylization strength, confirming that our approach enables fine-grained, post-inference control over the content–style trade-off. Together, these results show that our method supports both multi-style fusion and adjustable stylization without any additional optimization.

## 5 CONCLUSION

In this work, we propose Stylos, a feed-forward method for 3D stylization. By keeping geometry predictions on the self-attentive backbone path and conditioning appearance via global CrossBlocks on the style path, together with a voxel-space 3D style loss that aggregates multi-view features, Stylos achieves geometry-aware and view-consistent stylization. Our ablations show that the global CrossBlock for style injection better preserves geometric details than alternative style-content fusion modules. Extensive experiments across category-level and large-scale scene datasets demonstrate the generalization ability of our approach, achieving aesthetically pleasing stylization with strong cross-view consistency. In the future, we aim to scale Stylos to support higher-resolution inputs while improving efficiency, paving the way for practical 3D content creation.

# 6 ACKNOWLEDGEMENT

This research used both the DeltaAI advanced computing and data resource, which is supported by the National Science Foundation (award OAC 2320345) and the State of Illinois, and the Delta advanced computing and data resource which is supported by the National Science Foundation (award OAC 2005572) and the State of Illinois. Delta and DeltaAI are joint efforts of the University of Illinois Urbana-Champaign and its National Center for Supercomputing Applications. The allocations on Delta and DeltaAI [allocation number CIS250529] were made by the Advanced Cyberinfrastructure Coordination Ecosystem: Services & Support (ACCESS) program, which is supported by National Science Foundation grants #2138259, #2138286, #2138307, #2137603, and #2138296.

# 7 REPRODUCIBILITY STATEMENT

We have taken several steps to ensure the reproducibility of our work. The main paper describes all model architectures in Sec. 3. We provide detailed pseudo-code for the proposed 3D style loss function in Algo. 1, and will release the full implementation upon acceptance. All datasets used are publicly available, as described in Sec. 4. These resources should allow researchers to reproduce and extend our findings. In Appendix A.4, we clarify the training and inference details of the state-of-the-art methods for reproducible comparisons. In addition, we provide extensive qualitative results in Appendix A.6, and analyze efficiency trends with respect to the number of views per batch in Appendix A.2, where efficiency experiments are averaged over 100 iterations per setting.

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

## A  APPENDIX

### A.1  IMPLEMENTATION

**Implementation Details.**   For the geometry branch, we utilize VGGT as the backbone, comprising $L = 24$ Alternating-Attention Transformer layers. Both the geometry transformer and the depth DPT head are initialized using pretrained VGGT weights, while all remaining layers are initialized randomly. The style aggregator similarly employs 24 Transformer layers, configured with content-global self-attention followed by content-style cross-attention. For voxelization, each output of Geo Head associates a confidence value, which is converted into a normalized intra-voxel weight via softmax over Gaussians within the same voxel; these weights determine each Gaussian's contribution when aggregating voxel-level properties (opacity or color). The Gaussian Adapter maps the Geometry head and Style head predicted vector into explicit 3D Gaussian parameters. The module first splits the predicted vector into (i) isotropic scales, (ii) rotation quaternions, and (iii) degree-d spherical harmonic coefficients. Finally, we construct full covariance matrices using the predicted scale and rotation via the standard 3DGS covariance formulation. This yields the final set of Gaussian parameters: means, covariances, SH harmonics, opacities, scales, and normalized rotations.

In the geometry training stage, the entire network is pre-trained. The geometry outputs, poses and depth, are guided by a frozen VGGT teacher. The DINO encoder is trained to capture color and style features. For stylization fine-tuning, we optimize only the style aggregator and the style head. We set the voxel size to 0.002 for differentiable voxelization. For both geometry pre-training and stylization fine-tuning, we train the model using the AdamW optimizer for 15k iterations. We employ a cosine learning-rate schedule with a peak learning rate of $2 \times 10^{-4}$ and a 1k-iteration warmup.

**Data & Augmentation.**   Following Anysplat (Jiang et al., 2025), we randomly sample between 2 and 20 frames per clip, maintaining a fixed total of 20 frames per GPU. The input resolution is constrained to 448 pixels on the longest side, with the aspect ratio randomized between 0.5 and 1.0. Intrinsic augmentation is applied via random center-cropping (77%–100% of the original size) and random horizontal flipping. During Geometry Pretraining, we encourage the DINO encoder to better learn color and style tokens by occasionally replacing the content image with a style image and by applying color jitter to perturb its appearance, while keeping the original image as ground truth to supervise appearance-change learning.

**Loss Weights.**   Style weight $\lambda_{style} = 1.0$, content-reconstruction weight $\lambda_{cnt} = 0.1$, CLIP consistency weight $\lambda_{clip} = 1.0$, and total-variance regularization weight $\lambda_{tv} = 10.0$.

### A.2  EFFICIENCY SCALING

We further analyze the efficiency of Stylos by varying the number of views processed per batch. The results in Fig. 7 indicate a clear scaling pattern. For very small batches, such as 1–2 views, the average runtime per batch is low, around 0.3 s, but the measurements fluctuate heavily due

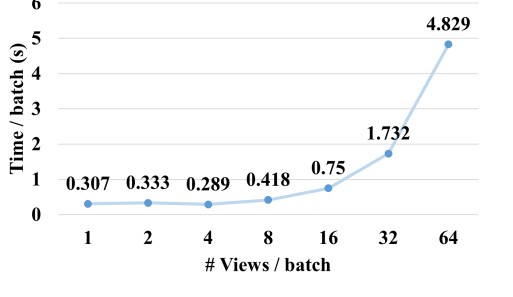

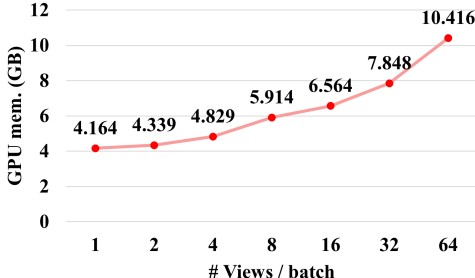

(a) Time per batch vs. # views per batch      (b) GPU memory usage vs. # views per batch

Figure 7: Scaling of inference time and GPU memory with the number of views per batch. Time is averaged over 100 iterations, and memory is reported as the peak allocation, without rendering cost.

to overheads and GPU scheduling. This makes single-view or extremely small-batch inference unstable, even though the raw latency is minimal. As the batch size increases, runtime grows roughly linearly, reaching 0.75 s at 16 views and 4.83 s at 64 views. Importantly, once the batch size exceeds 4–8 views, the runtime becomes much more stable, showing only minor variation across repeated runs. Memory usage shows a similarly consistent trend: GPU memory rises steadily from 4.16 GB at 1 view to 10.42 GB at 64 views. The growth is close to linear with respect to batch size, which makes resource requirements easy to estimate when scaling to larger workloads.

### A.3 ABLATION ANALYSIS

**Content-Style Coupling.** Stylos consists of a style pathway which contributes to the color information for the predicted 3DGS representation, and a content pathway for the other 3DGS parameters. In the style pathway, we introduce a style aggregator with cross-attention between self-attention and MLP to entangle the content and style tokens. Our training pipeline consists of two stages. The first stage focuses on 3D scene reconstruction, where a random content view is used as the style reference; Stage 2 performs actual stylization using a real artistic style image. When designing Stylos, it is therefore essential to verify that introducing the style-content fusion blocks does not harm the reconstruction process.

So, we compare three CrossBlock designs (global, frame, and both) by calculating PSNR, SSIM and LPIPS between the rendered multi-view rgbs and their corresponding ground-thruth after Stage 1 training, using the first content view as the style reference. As shown in the Table 1, the global-only variant achieves the best reconstruction accuracy. This behavior is expected, because each view attends to the style tokens independently without interacting with the other views in frame attention. Treating a single view as the style reference for all views forces each view to match its patch-level representation to the same reference individually, which distorts the rendered images. Fig. 2 shows that the frame attention significantly blurs the pizza box and toppings.

**Image-, Scene-, and Voxel-level Style Losses.** In this section, we provide additional visual comparisons showing that the proposed scene and 3D style losses significantly outperform the baseline image style loss. Specifically, we observe that the scene- and 3D-level losses produce cleaner and more coherent textures, such as the wooden floor in Fig. 8. As shown in Fig. 9, the 3D style loss is also able to stylize a larger portion of the table surface compared with the image loss. In Fig. 10, the seams on the tablecloth are well preserved under the 3D loss, while they become partially distorted when using the image or scene losses. We will release our implementations of all three style losses to facilitate further exploration by the community.

### A.4 REPRODUCIBLE EXPERIMENTS

**Reproduce State-of-the-art Models.** For StyleGaussian (Liu et al., 2024), SGSST (Galerne et al., 2025), StylizedGS (Zhang et al., 2025), and GStyle (Kovács et al., 2024), we use their officially released codebases and pretrained weights. We either run inference directly with the provided models or follow the authors' instructions to train the models when necessary. All of them are evaluated on GH200 GPUs for fair comparisons.

We notice that StylizedGS (Zhang et al., 2025) fail in serveral style cases, producing stylized images with pure colors with barely visible geomtry details. It significantly improves its consistency scores in our experiments. For fair comparisons, we exclude these failure cases when calculating consistency and ArtScore for StylizedGS, as they could lead to very small values.

**Stylization Quality Comparisons.** In Table 3 and 4, * denotes that failure stylization cases for StylizedGS* (Zhang et al., 2025) are excluded for fair comparisons. Please refer to our released codes for additional visual examples. Table 5 and Table 6 are provided as supplementary quantitative results to Table 4.

**Stylization Time Comparisons.** We run 48 samples as a hardware warm-up phase, and calculate the average inference speeds over the following 112 samples. For fair comparisons, all methods are evaluated on the same hardware, using an NVIDIA GH200 GPU with identical configurations. The reported stylization time of the per-scene fitting required method considers per-scene training time

Table 5: Following StyleID (Chung et al., 2024), we additionally report sylization quality metrics, FID, LPIPS, LPIPS-gray, CFSD, color matching loss (HistoGAN loos), as supplementary to Table 4.

| Method | Train | | | | | Truck | | | | |
|---|---|---|---|---|---|---|---|---|---|---|
| | FID↓ | LPIPS↓ | Gray↓ | CFSD↓ | CM Loss↓ | FID↓ | LPIPS↓ | Gray↓ | CFSD↓ | CM Loss↓ |
| StyleGaussian | 34.59 | 0.483 | 0.377 | 0.190 | 0.418 | 28.53 | 0.522 | 0.367 | 0.131 | 0.382 |
| G-Style | **14.80** | **0.471** | **0.364** | **0.165** | **0.185** | **13.65** | **0.512** | **0.353** | 0.087 | **0.176** |
| StylizedGS | 22.90 | 0.707 | 0.648 | 0.216 | 0.396 | 22.65 | 0.779 | 0.713 | **0.084** | 0.417 |
| SGSST | 24.15 | 0.520 | 0.409 | 0.220 | 0.257 | 19.50 | 0.577 | 0.445 | 0.177 | 0.196 |
| Styl3R | 19.77 | 0.670 | 0.598 | 0.244 | 0.364 | 19.28 | 0.682 | 0.575 | 0.102 | 0.350 |
| *Stylos* (ours) | 15.30 | 0.620 | 0.529 | 0.223 | 0.241 | 16.67 | 0.625 | 0.471 | **0.084** | 0.237 |

Table 6: Following StyleID (Chung et al., 2024), we additionally report sylization quality metrics, FID, LPIPS, LPIPS-gray, CFSD, color matching loss (HistoGAN loos), as supplementary to Table 4.

| Method | M60 | | | | | Garden | | | | |
|---|---|---|---|---|---|---|---|---|---|---|
| | FID↓ | LPIPS↓ | Gray↓ | CFSD↓ | CM Loss↓ | FID↓ | LPIPS↓ | Gray↓ | CFSD↓ | CM Loss↓ |
| StyleGaussian | 30.54 | 0.506 | 0.413 | 0.138 | 0.467 | 25.23 | 0.569 | 0.454 | 0.189 | 0.480 |
| G-Style | **13.93** | **0.498** | **0.395** | **0.092** | **0.208** | **16.17** | **0.500** | **0.364** | 0.103 | **0.179** |
| StylizedGS | 28.33 | 0.815 | 0.728 | 0.102 | 0.443 | 33.73 | 0.876 | 0.827 | **0.074** | 0.570 |
| SGSST | 23.95 | 0.552 | 0.458 | 0.204 | 0.264 | 20.93 | 0.529 | 0.415 | 0.233 | 0.228 |
| Styl3R | 17.14 | 0.646 | 0.573 | 0.124 | 0.314 | 22.38 | 0.637 | 0.556 | 0.097 | 0.335 |
| *Stylos* (ours) | 16.61 | 0.558 | 0.457 | 0.098 | 0.252 | 16.26 | 0.625 | 0.509 | 0.080 | 0.242 |

besides the rendering time. We strictly follow the original authors' training setups and inference pipelines. All 3DGS-based methods (Liu et al., 2024; Galerne et al., 2025; Zhang et al., 2025; Kovács et al., 2024) are evaluated on the full-resolution images, whereas single-forward methods, including Styl3R (Wang et al., 2025b) and Stylos, are evaluated at their preset input resolutions, i.e., $256 \times 256$ and $448 \times 448$, respectively.

## A.5 LIMITATIONS

*Stylos* tends to underperform on scenes dominated by high-frequency or highly cluttered structures, such as dense foliage, thin branches, or wire-like elements, where preserving fine geometric detail is particularly challenging. The method also struggles with style categories that introduce strong global lighting shifts or extreme color palettes, which can lead to over-saturation or loss of subtle appearance cues. In addition, performance decreases as the number of input views increases. This degradation primarily stems from limitations of the VGGT backbone: as view count grows, its underlying geometric reconstruction becomes less stable, which in turn negatively affects the quality of the stylized outputs. More specifically, the failure cases can be grouped into three categories: (1) Failures caused by inaccurate scene reconstruction, such as missing geometric details or incorrect region colors; (2) Failures caused by generating colors that are absent from the original style images. (3) Failures caused by introducing blurry or over-smoothed style colors.

## A.6 ADDITIONAL VISUAL RESULTS

Figure 14-15 exhibit all the 50 test style images for quantitative comparisons. Figure 16-25 presents additional qualitative results of Stylos on diverse style–content pairs. Across different styles and object categories, Stylos produces visually pleasing stylizations that respect the input content geometry while transferring the target artistic appearance. These examples further demonstrate the versatility and robustness of our approach beyond the main results in the paper.

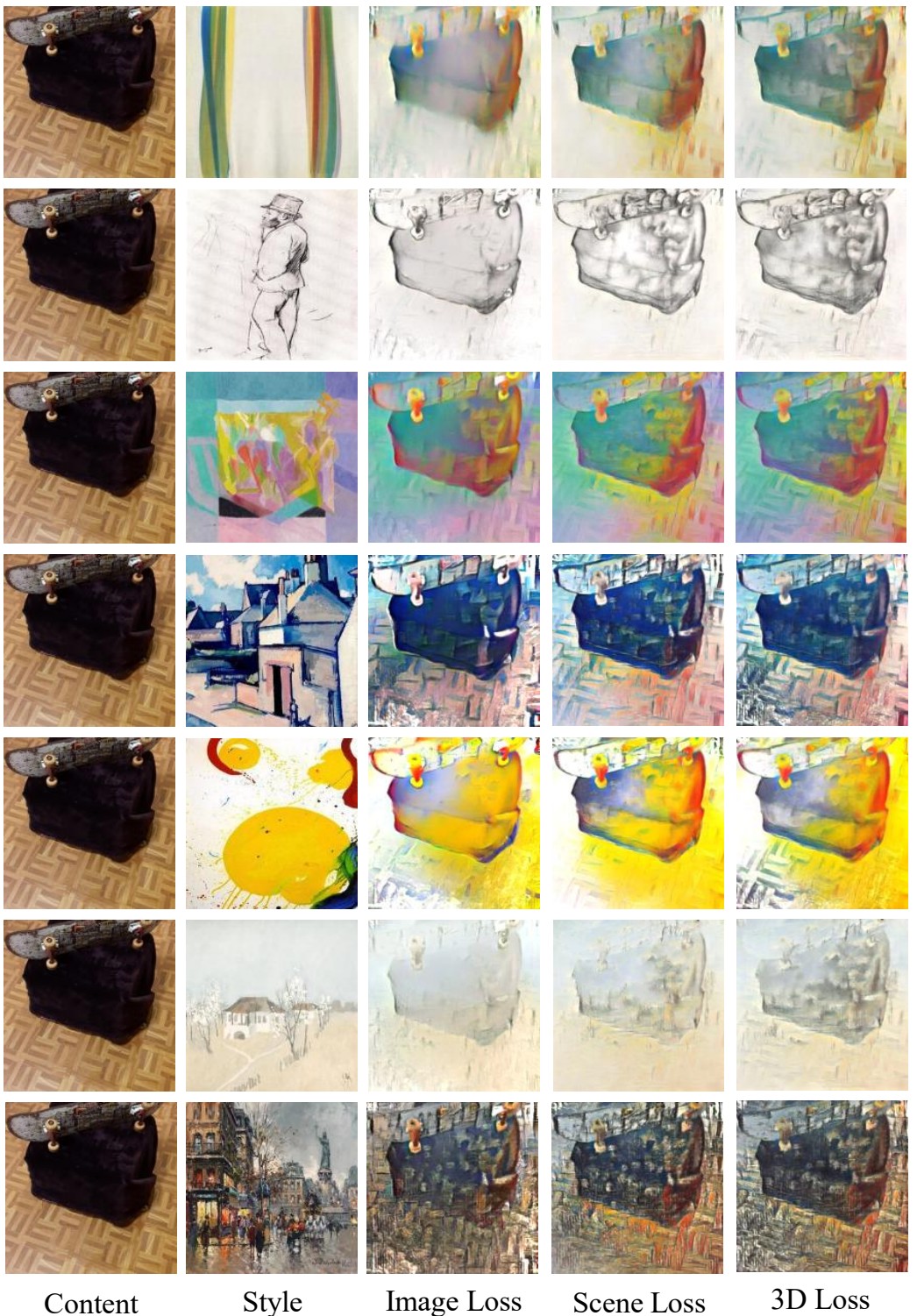

Content      Style      Image Loss      Scene Loss      3D Loss

Figure 8: Visual comparisons of different style losses on the CO3D skateboard scene. It is clear that the 3D style loss provides a stronger sense of 3D geometry and cleaner wooden floor textures evaluated on varying style images.

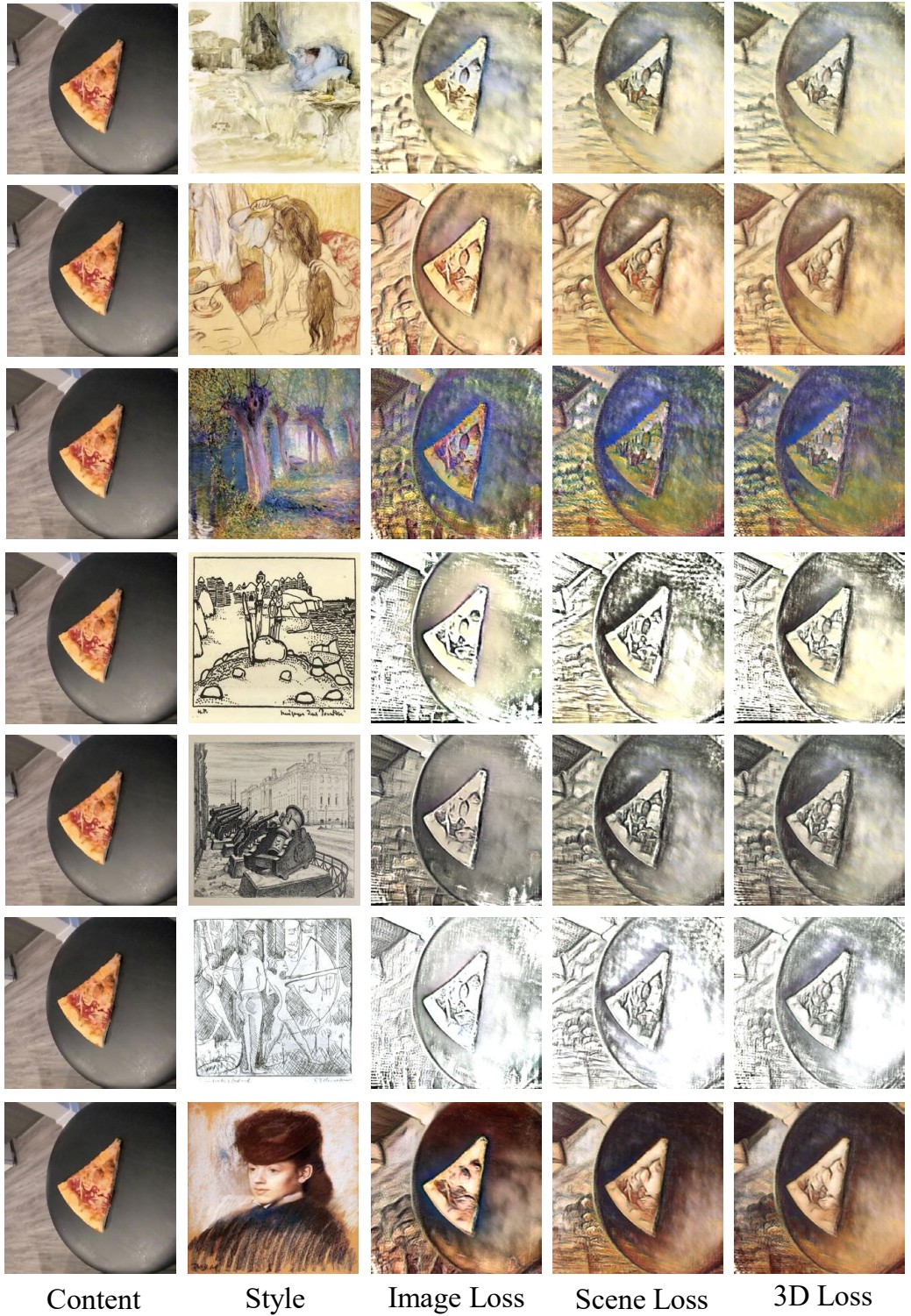

Content   Style   Image Loss   Scene Loss   3D Loss

Figure 9: Visual comparisons of different style losses on the CO3D pizza scene. It is clear that the 3D style loss provides stronger artistic stylization to the table surface, while the image style loss often fails to impose the expected styles to cover the whole table surface.

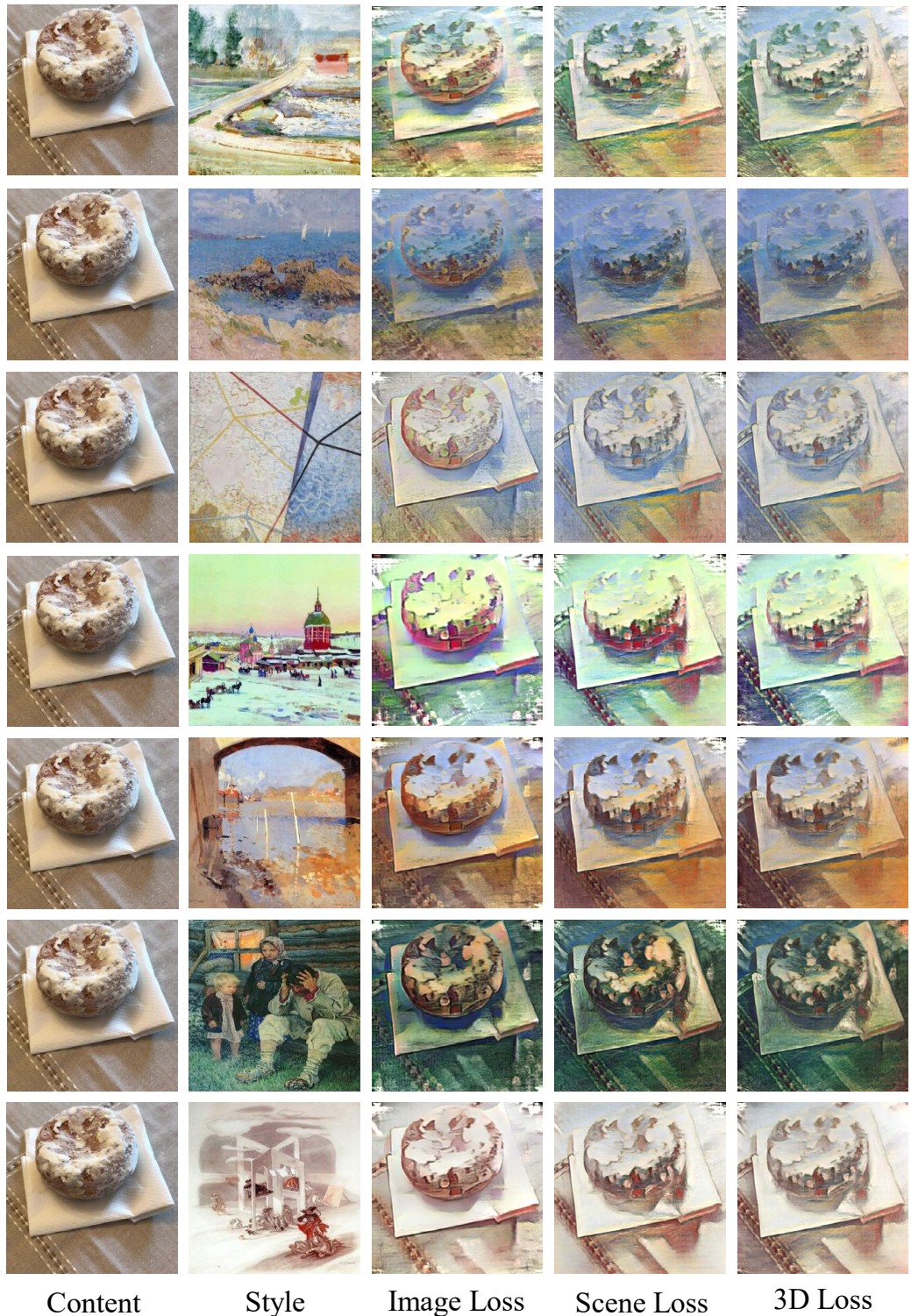

Content        Style        Image Loss        Scene Loss        3D Loss

Figure 10: Visual comparisons of different style losses on the CO3D donut scene. It is clear that the 3D style loss provides faithful stylization to the table cover while preserving the seams on the tablecloth compared to both the image style loss and the scene style loss.

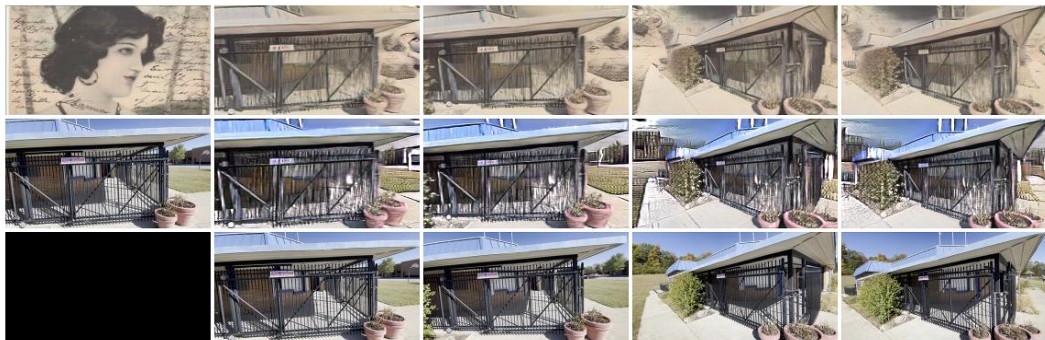

(a) The reconstructed metal bars on the gate are unclear as shown in the mid row, and the stylized metal bars at the top row are barely visible either.

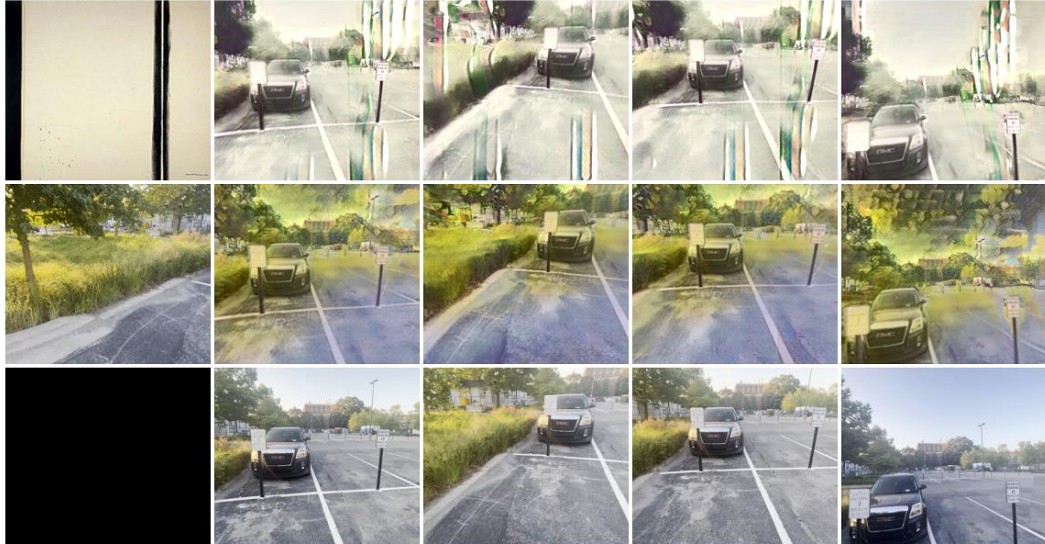

(b) The green colors fail to correspond to the intended regions in the reconstructed scene (at the mid row), and the stylized multi-view images mistakenly interpret the green area as part of the target pattern, leading the model to apply stylization to it, (at the top row).

Figure 11: Failures caused by inaccurate scene reconstruction, such as missing geometric details or incorrect region colors. Top row: The input style image and the stylized multi-view results; mid row: a reference frame from the input contents and the reconstructed multi-view images; bottom row: the input contents.

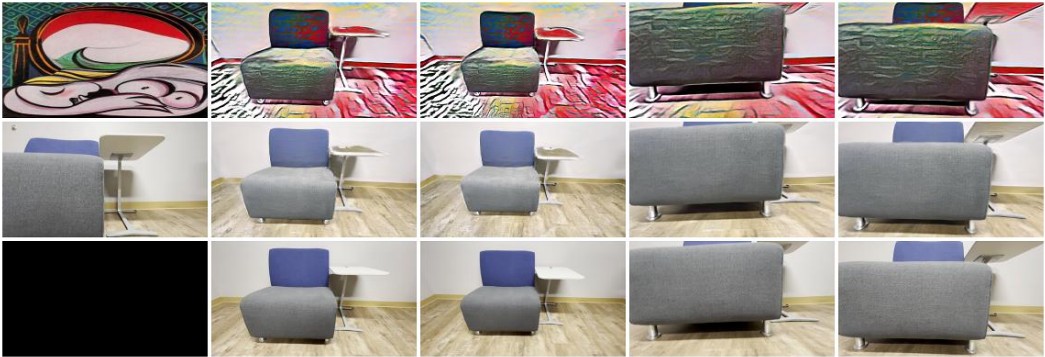

(a) The reconstructed scene is generally accurate. However, our model introduces a pure white region in the bottom-left area (at the top row), which does not appear in the original style image.

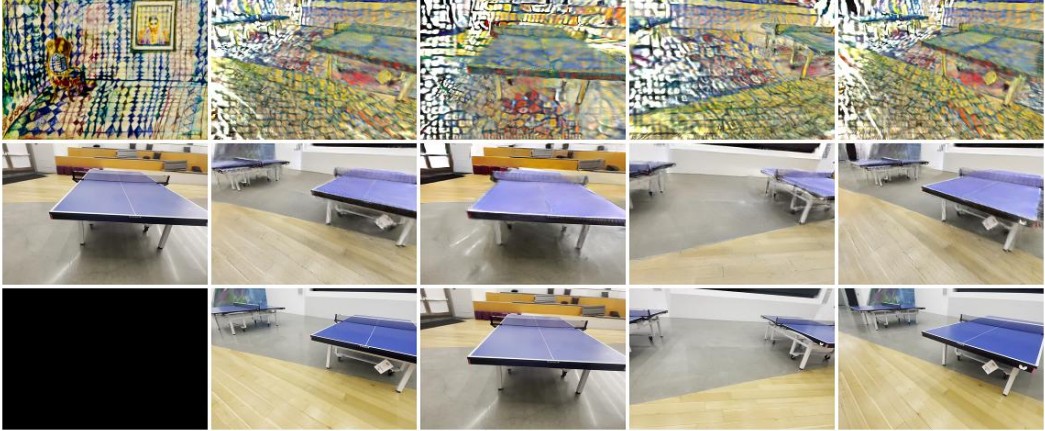

(b) The reconstructed scene is generally accurate. However, our model hallucinates a pure white region on the left side (at the top row), which is absent from the original style image.

Figure 12: Failures caused by generating colors that are absent from the original style images. Top row: The input style image and the stylized multi-view results; mid row: a reference frame from the input contents and the reconstructed multi-view images; bottom row: the input contents.

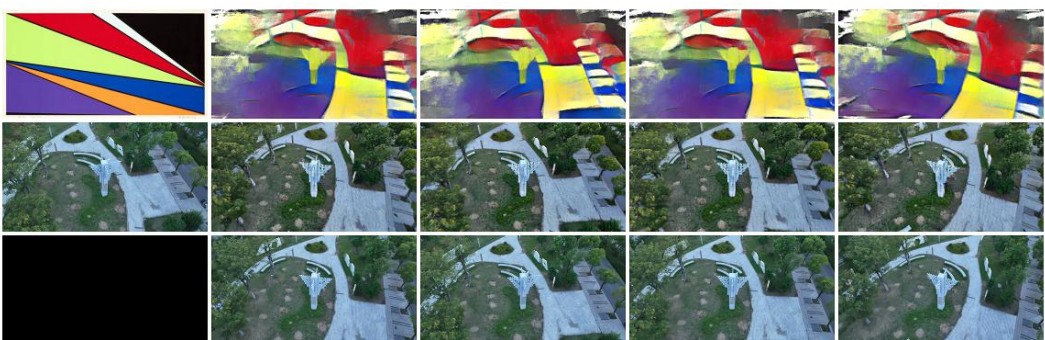

(a) Our model produces a blurred stylized region in the upper-left area of the results (as shown in the top row).

Figure 13: Failures caused by introducing blurry or over-smoothed style colors. Top row: The input style image and the stylized multi-view results; mid row: a reference frame from the input contents and the reconstructed multi-view images; bottom row: the input contents.

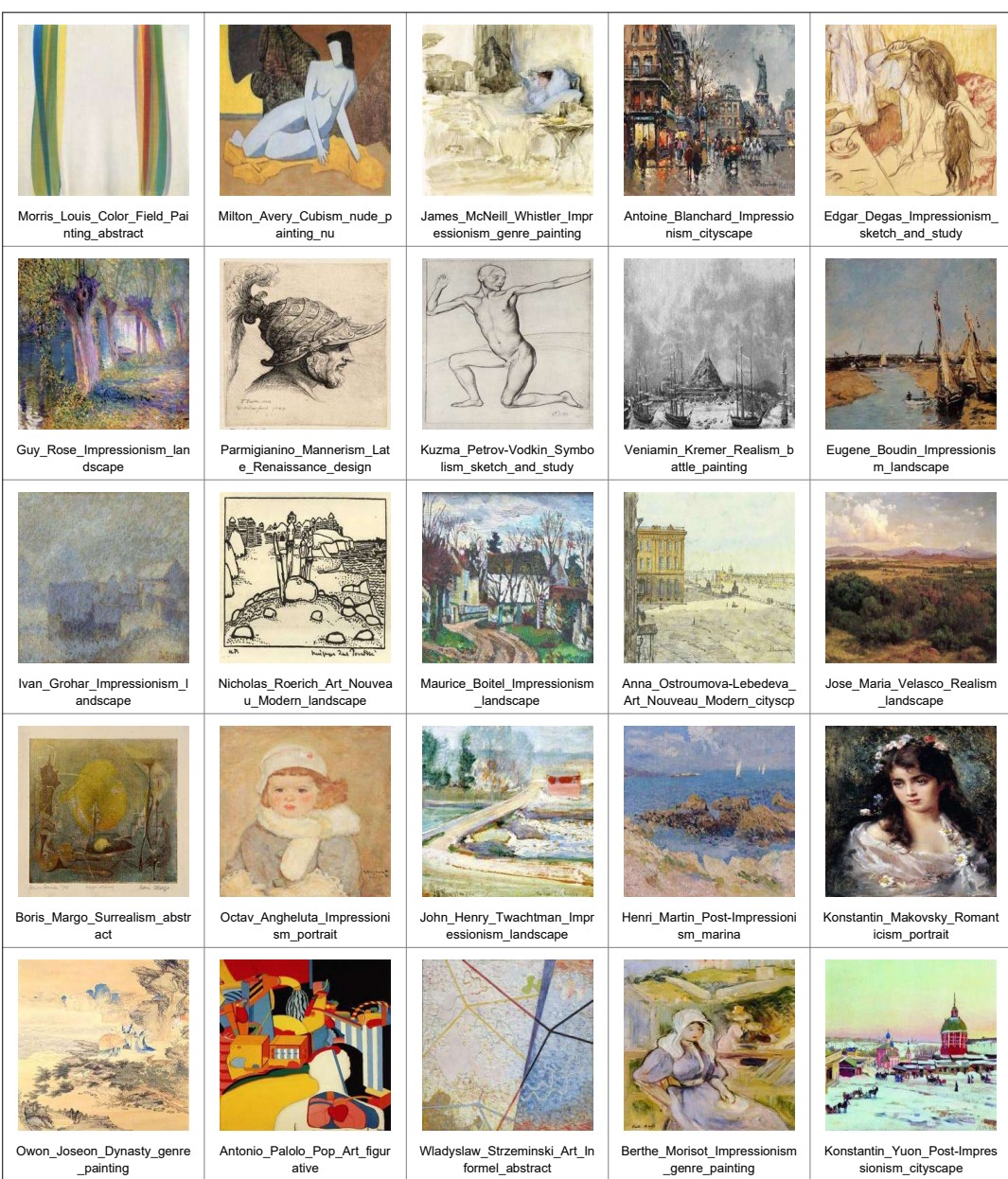

Figure 14: Style images from WikiArt (2025) used for model evaluation in all quantitative comparisons, while not used for training Stylos.

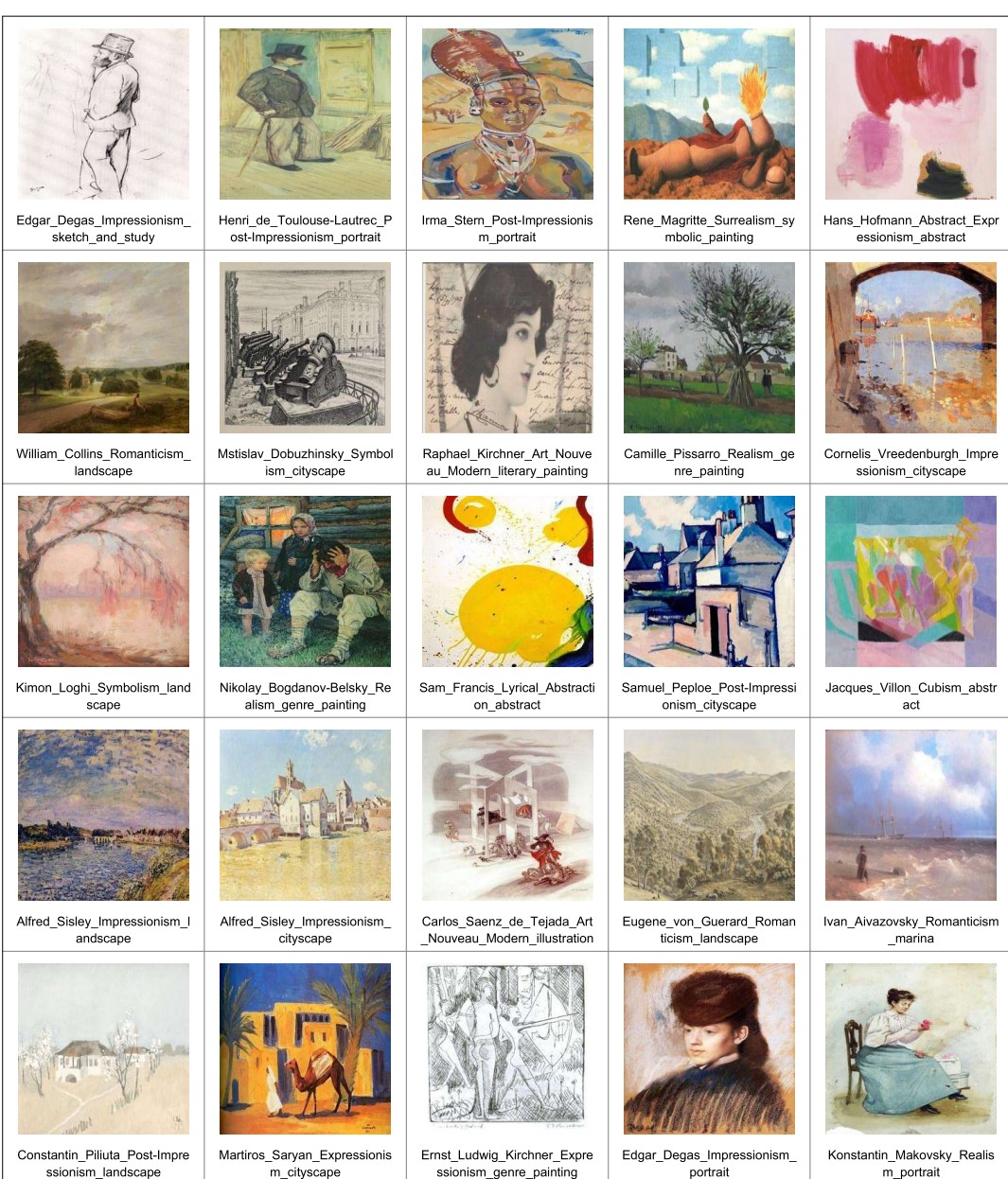

Figure 15: Style images from WikiArt (2025) used for model evaluation in all quantitative comparisons, while not used for training Stylos.

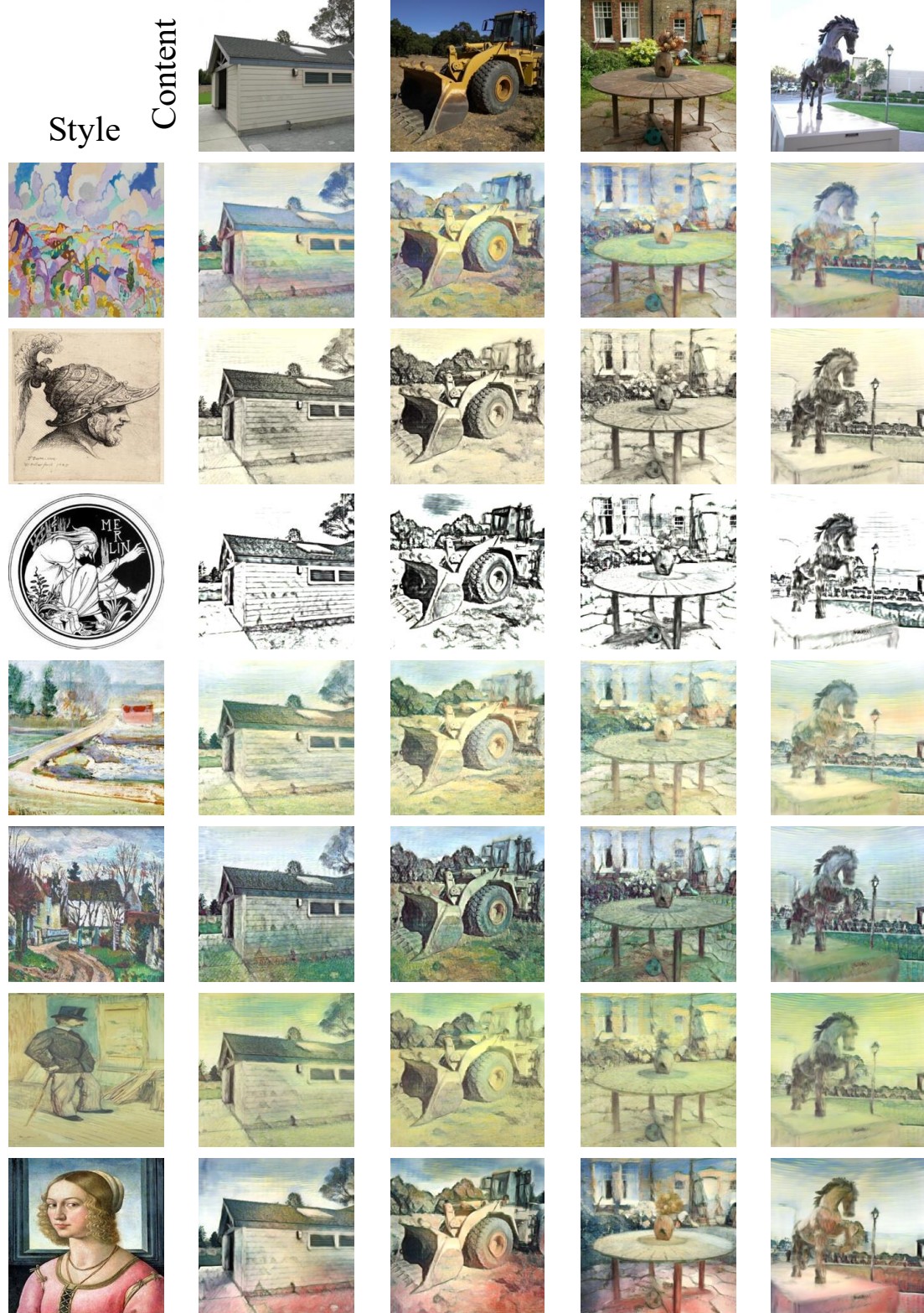

Figure 16: Additional visual results of Stylos on diverse style–content pairs on Tanks and Temples Dataset (Knapitsch et al., 2017). Stylos consistently produces visually pleasing stylizations that preserve content geometry while transferring the target style.

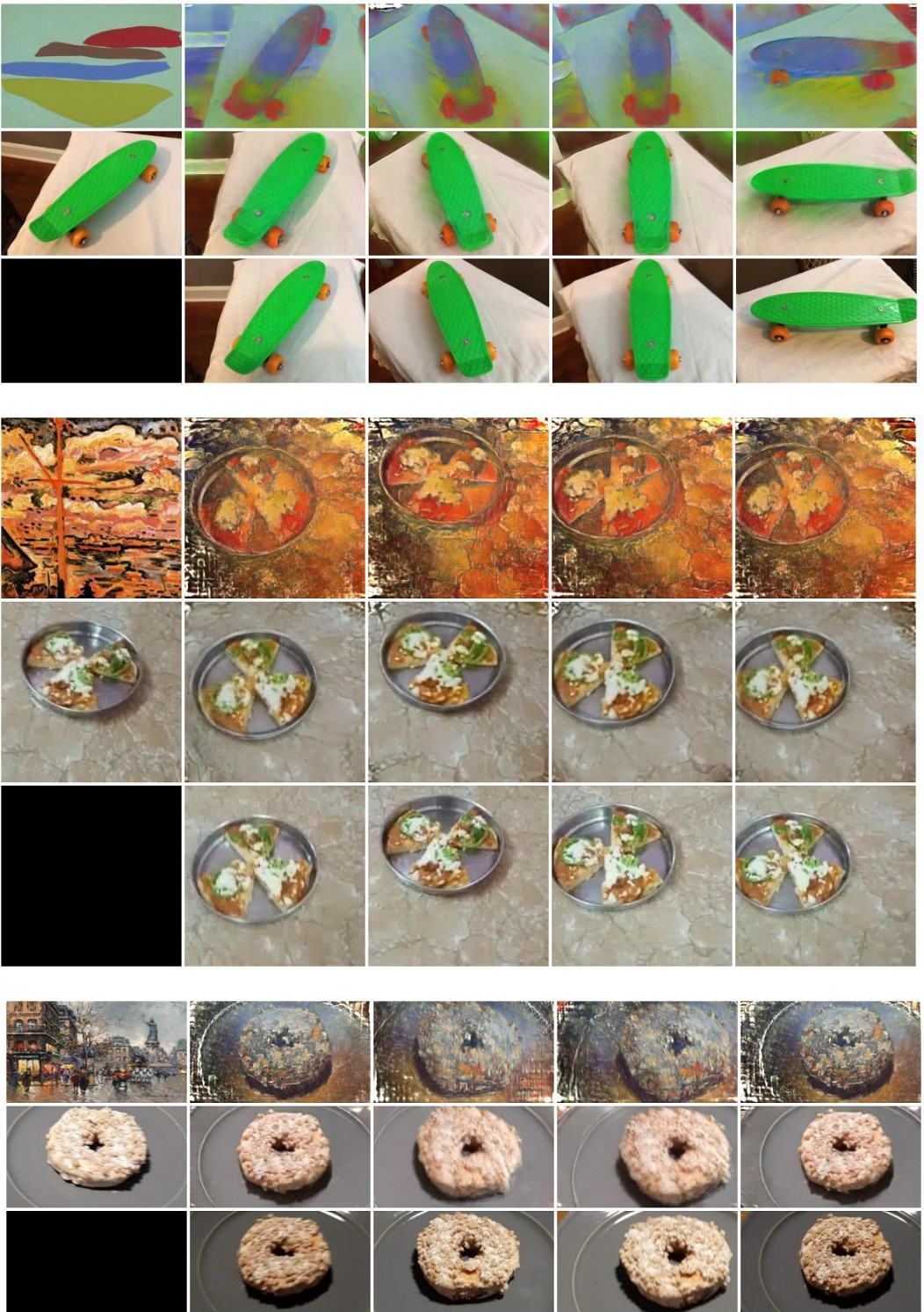

Figure 17: Additional visual results of Stylos on the CO3D dataset. The top left is the style image while the rest of the top row are the rendered images. The mid row are the rendered images conditioned on the mid left imamge. The bottom row is the content inputs.

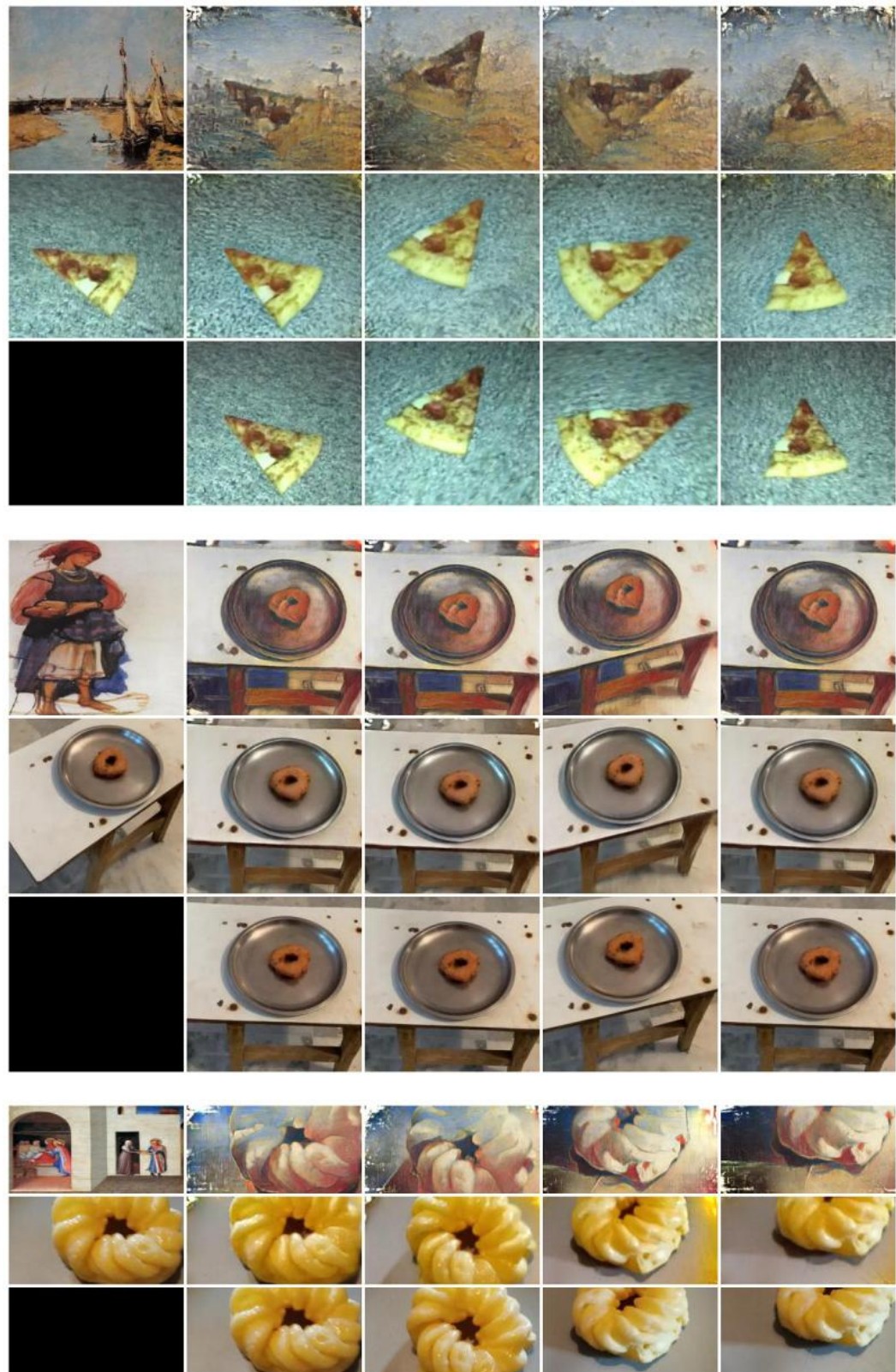

Figure 18: Additional visual results of Stylos on diverse style–content pairs.

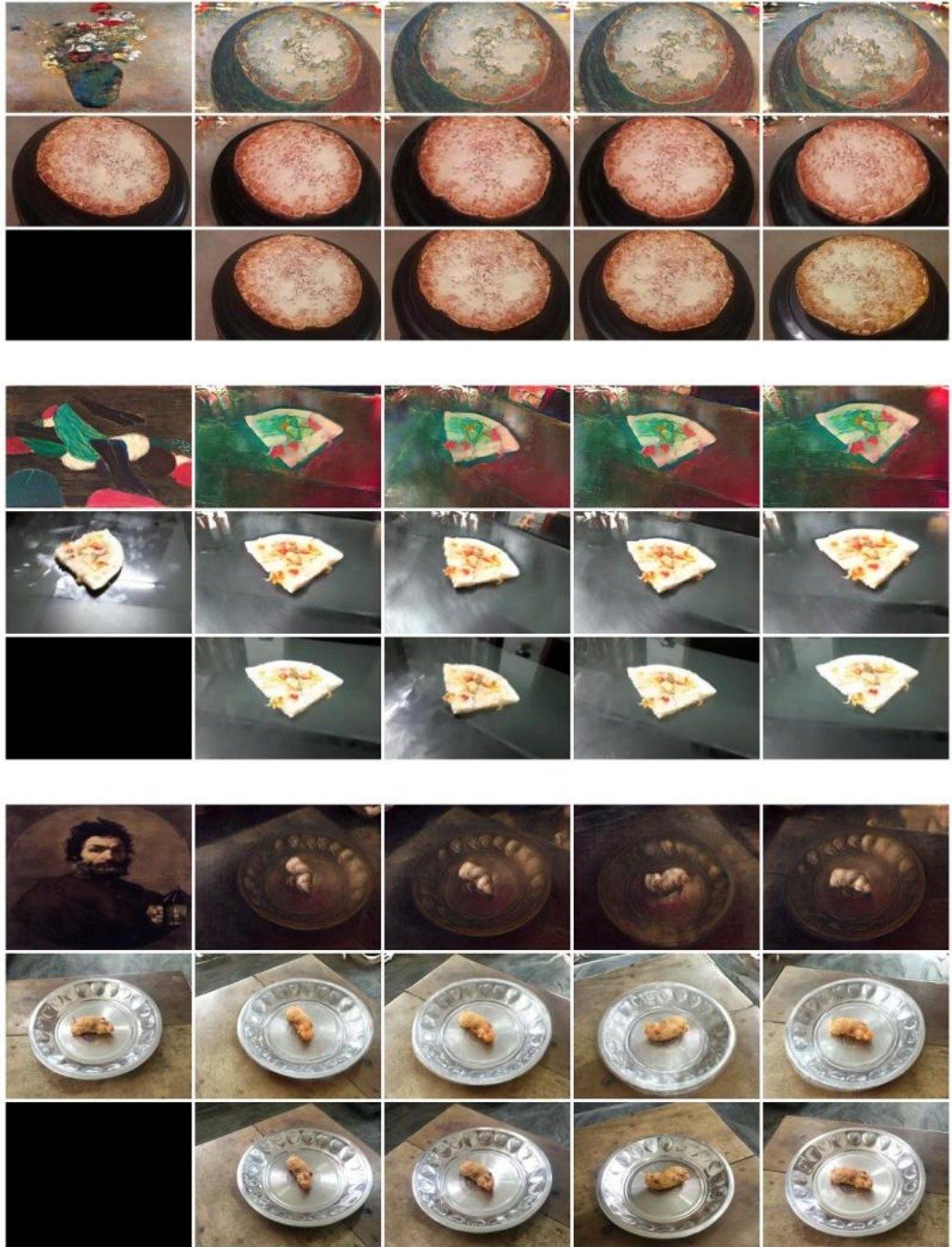

Figure 19: Additional visual results of Stylos on diverse style–content pairs.

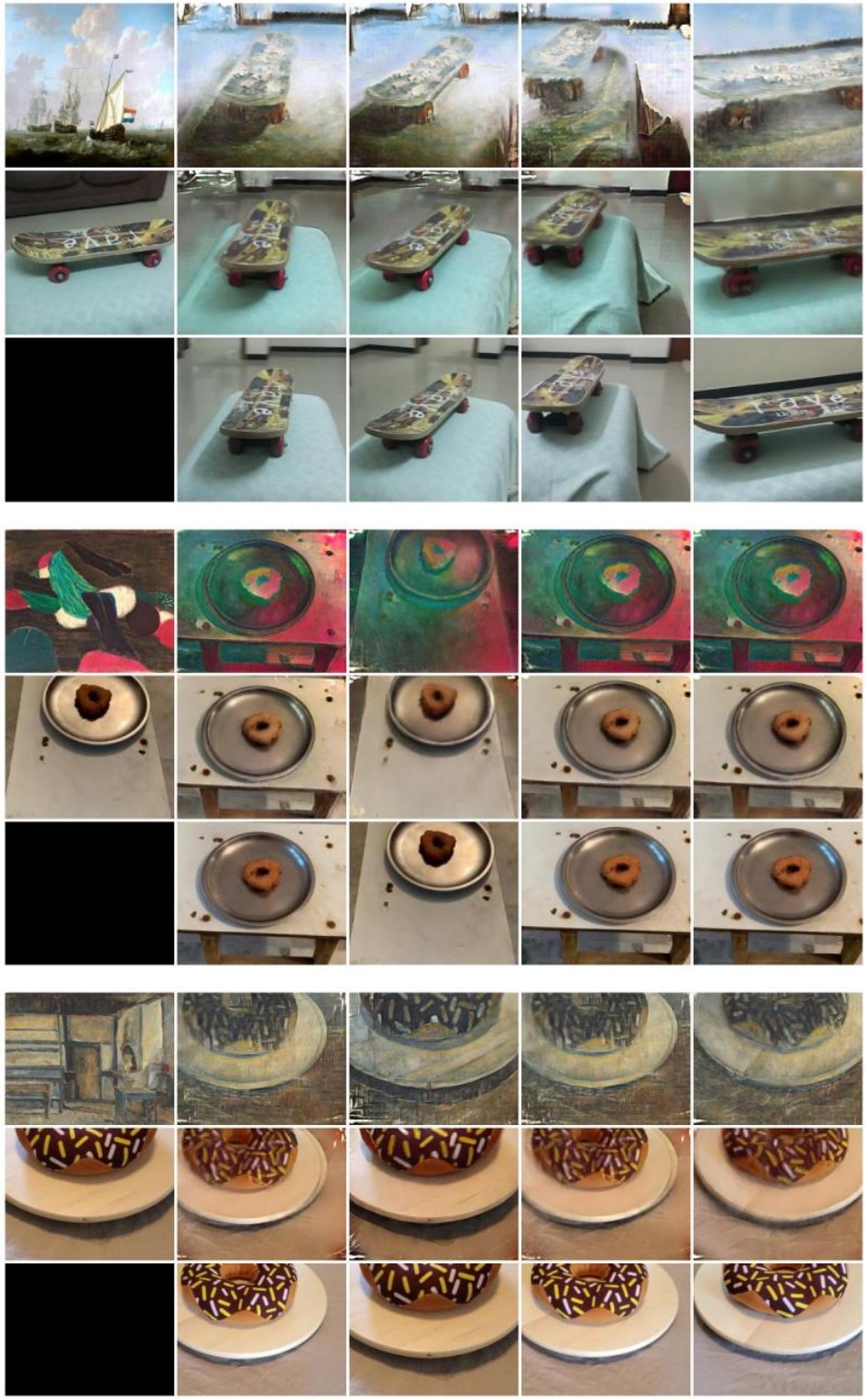

Figure 20: Additional visual results of Stylos on diverse style–content pairs.

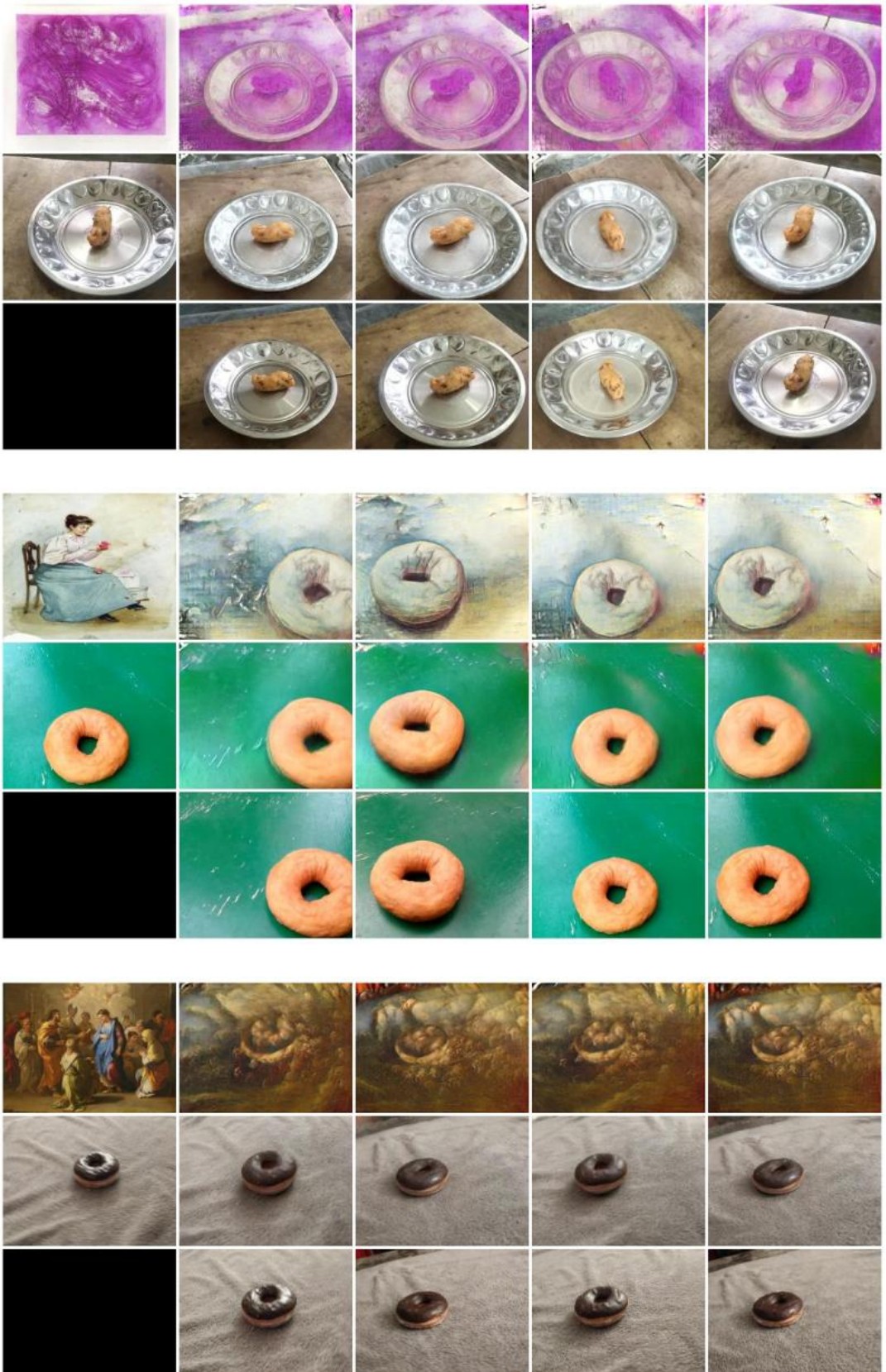

Figure 21: Additional visual results of Stylos on diverse style–content pairs.

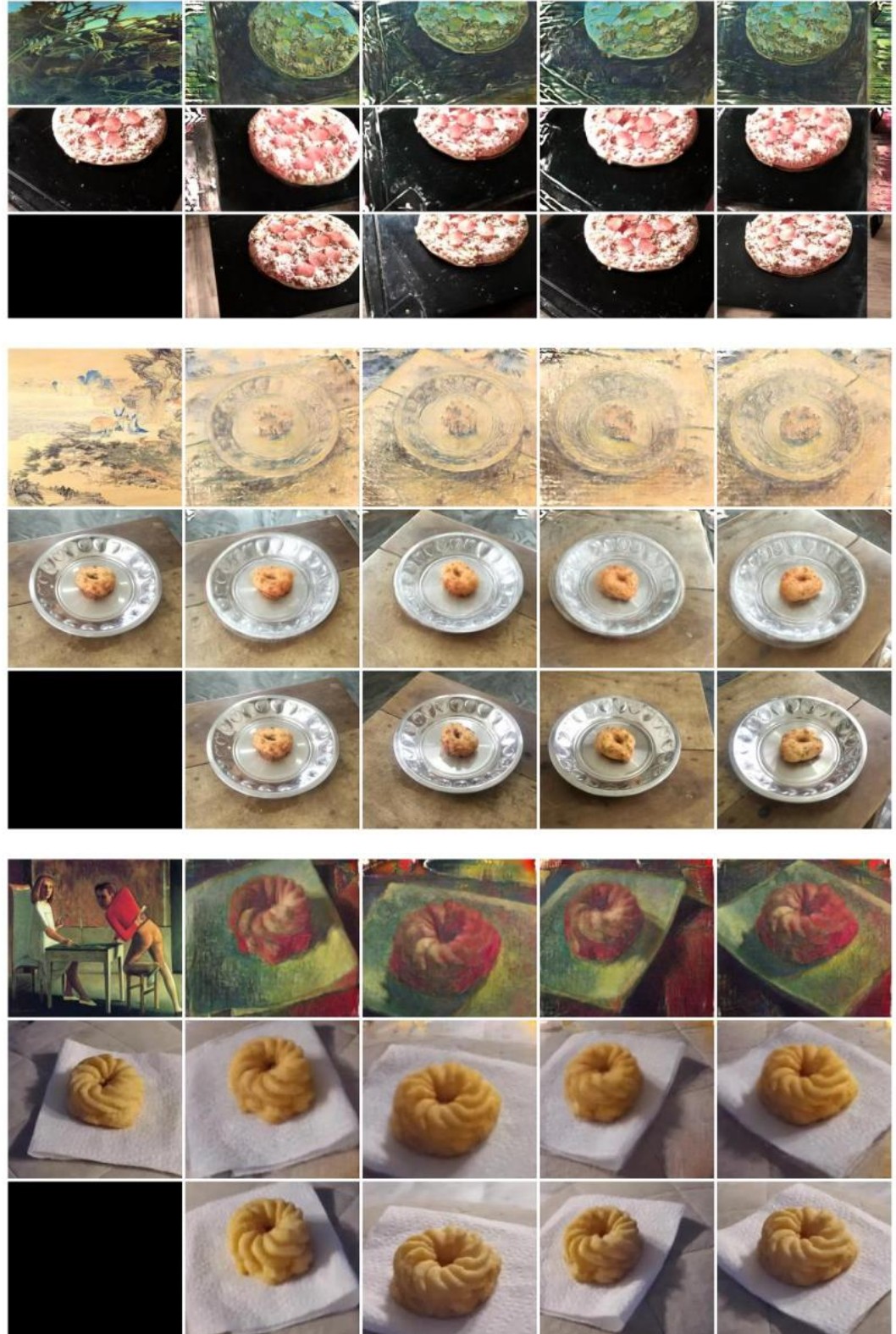

Figure 22: Additional visual results of Stylos on diverse style–content pairs.

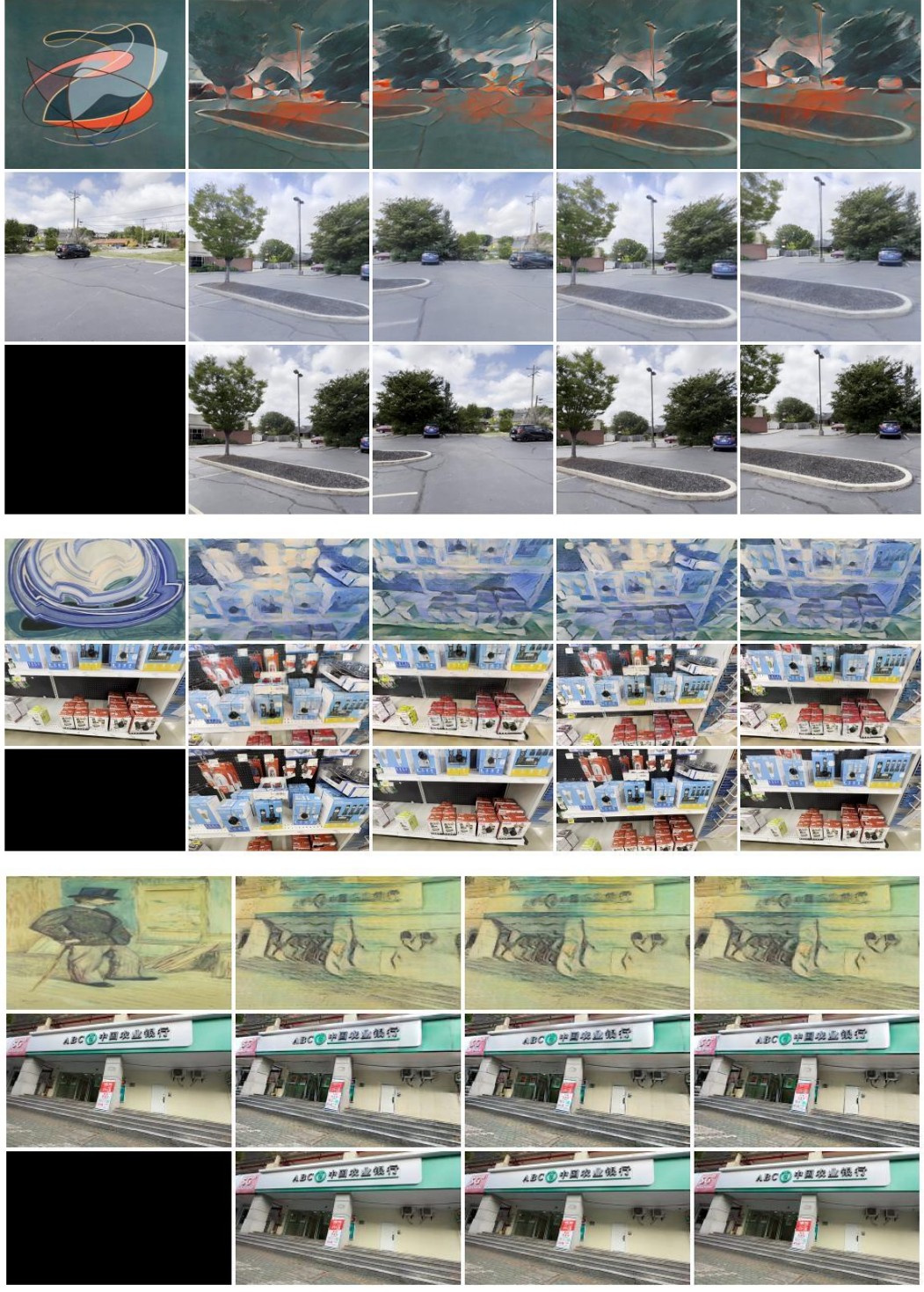

Figure 23: Additional visual results of Stylos on the DL3DV-10K dataset. The top left is the style image while the rest of the top row are the rendered images. The mid row are the rendered images conditioned on the mid left imamge. The bottom row is the content inputs.

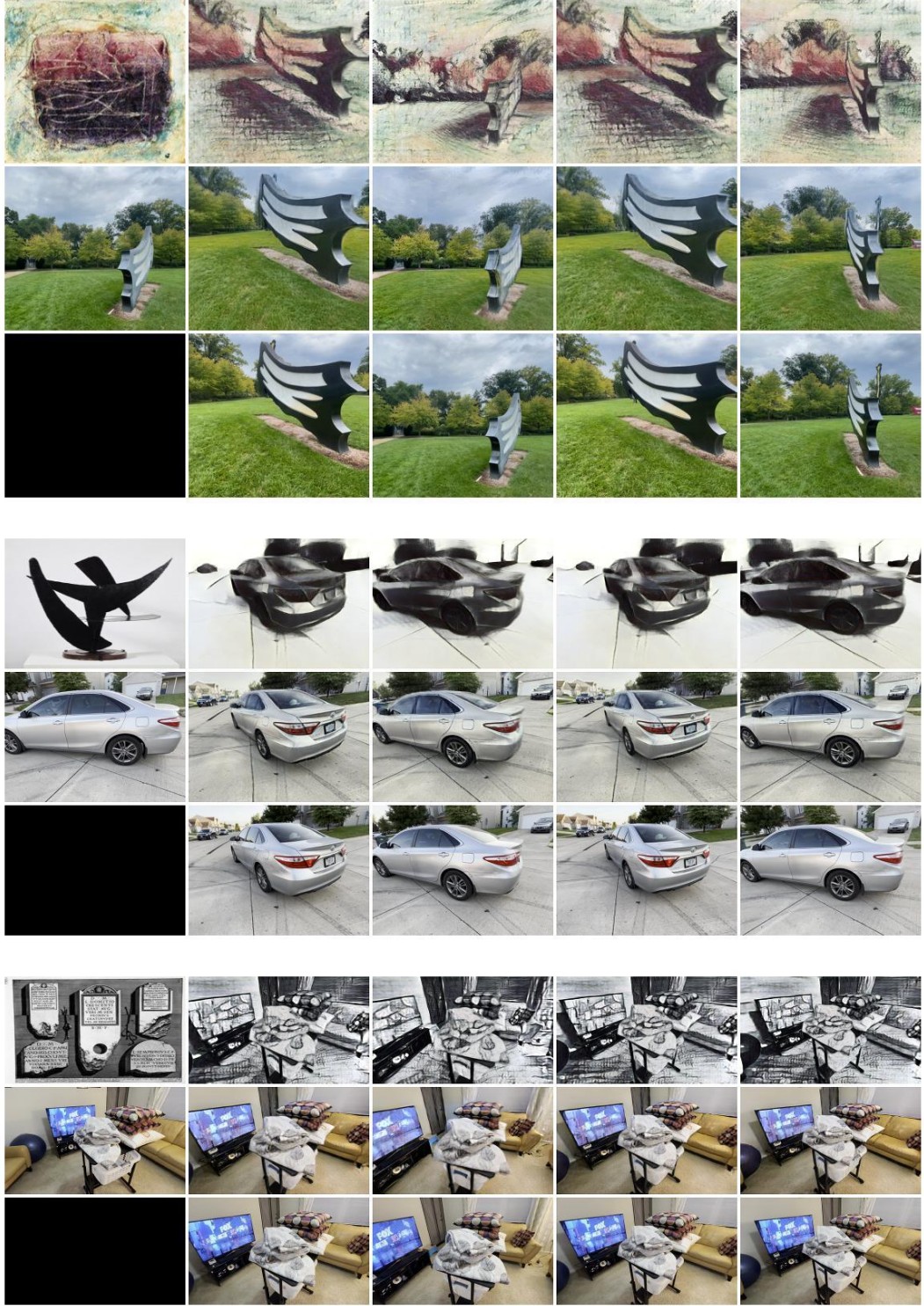

Figure 24: Additional visual results of Stylos on diverse style–content pairs.

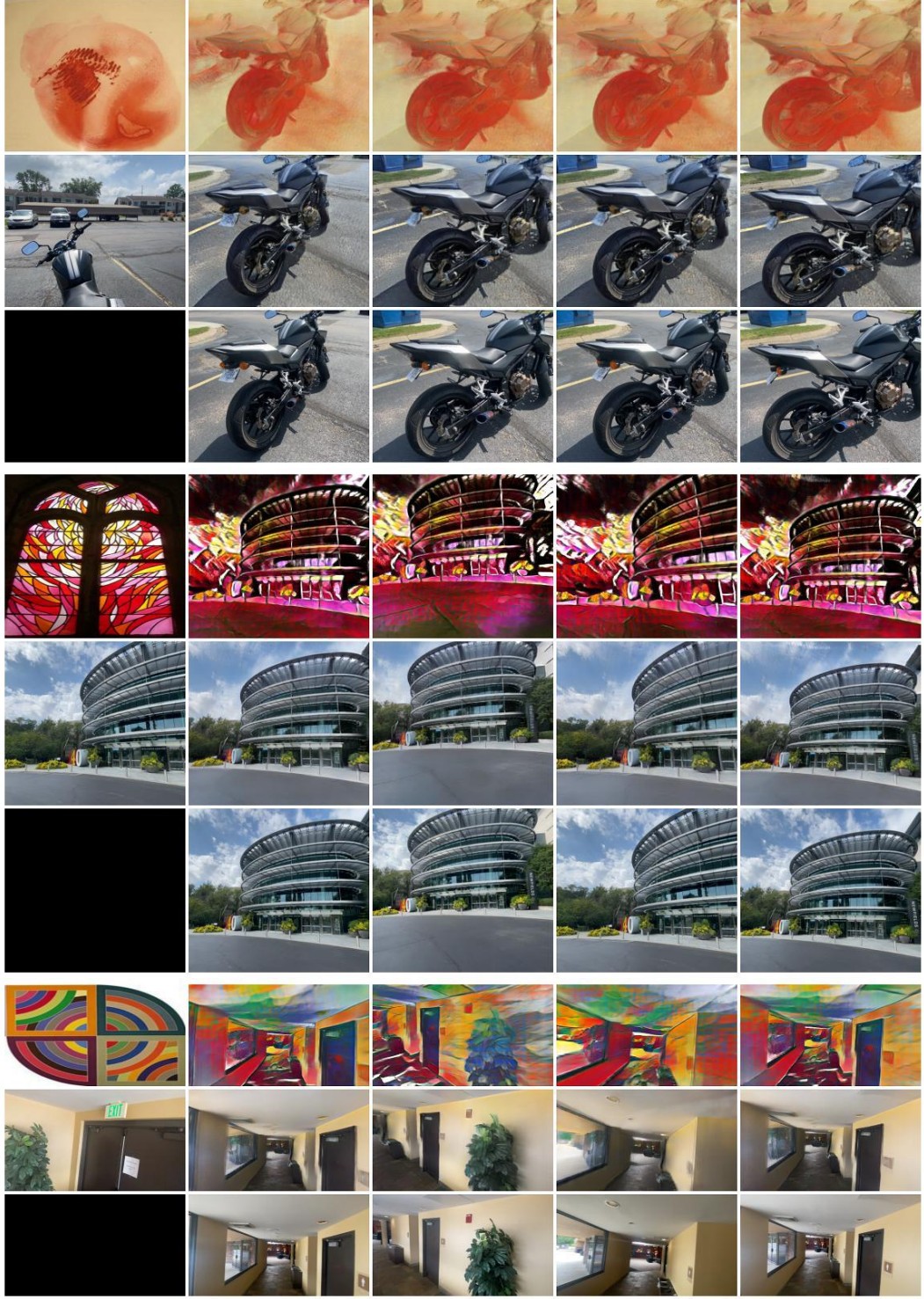

Figure 25: Additional visual results of Stylos on diverse style–content pairs.

