# OpenReview forum: "Stylos: Multi-View 3D Stylization with Single-Forward Gaussian Splatting"
_ICLR.cc/2026/Conference — ICLR 2026 Poster_

### Official Review · Reviewer_LtqC · 2025-10-27

**Soundness:** 3
**Presentation:** 3
**Contribution:** 3
**Rating:** 8
**Confidence:** 3

**Summary:**

The paper proposes Stylos, a feed-forward 3D Gaussian stylization framework. Given one or more unposed RGB views of a scene plus a single style reference image, Stylos directly predicts a stylized 3D Gaussian scene in a single forward pass, without per-scene optimization or known camera intrinsics/extrinsics. Stylos explicitly separates geometry and style: geometry is inferred by a VGGT-like alternating-attention backbone, while style is injected through a Style Aggregator that applies global cross-attention from the style tokens to all content-view tokens. The paper also introduces a voxel-based 3D style loss: instead of matching style statistics (mean/variance in VGG feature space) per-frame, Stylos fuses multi-view features into a shared voxel grid and aligns those 3D features to the style distribution, which the authors argue enforces cross-view consistency and preserves geometry. Experiments on CO3D, DL3DV-10K, and Tanks and Temples report improved multi-view consistency (LPIPS/RMSE) and competitive perceptual/artistic quality (ArtScore) compared to per-scene stylization baselines such as StyleRF and StyleGS, and include ablations on cross-attention design and style loss variants.

**Strengths:**

- Practical significance. Single-forward stylization of an entire 3D Gaussian scene (poses + Gaussians + colors) without per-scene finetuning or known camera parameters could be transformative for content pipelines. This is not just incremental NeRF/3DGS stylization but closer to the real situation, like “instant turn-this-video-into-stylized-3D-assets.”
- Architectural clarity. Clean separation between a geometry backbone and a Style Aggregator branch (global cross-attention conditioning on the style image). The ablation across “frame-only,” “hybrid,” and “global-only” variants is convincing and shows why global conditioning improves structure and texture coherence.
- Breadth of evaluation. The paper reports not only reconstruction-like metrics (PSNR/SSIM/LPIPS) but also temporal / multi-view consistency metrics (LPIPS & RMSE at different frame gaps) and ArtScore for perceptual “artness,” and demonstrates cross-category and cross-scene generalization.

**Weaknesses:**

- 3D style loss justification is empirical. The 3D voxel-statistics loss gives better LPIPS/RMSE consistency, but the paper does not analyze potential side effects such as global color bleeding or loss of local style detail in poorly observed regions. Some intuition or failure-case visualization would strengthen the argument that voxel-statistics matching is fundamentally better than 2D scene-level matching.
- Scalability limits. Although Stylos is advertised as handling up to “hundreds of views,” the scaling experiment shows visible degradation beyond ~32 views per batch (edge artifacts, instability in the Gaussian representation). The method may still be quite useful, but this practical constraint, probably brought by VGGT, should be communicated more prominently.

**Questions:**

- Failure cases of the 3D style loss.
Are there any visual failure cases where voxelized style statistics hurt local detail (e.g., oversmoothing, texture bleeding in occluded regions)? Right now we only see success cases and quantitative gains.

---

> ### Author Response · Authors · 2025-11-25
> **Response to Reviewer LtqC, Part I**
>
> We sincerely appreciate the reviewer for giving our work a positive rating and providing several insightful comments. We have carefully revised the paper and added new experimental results to address the reviewer’s questions. Our responses are as follows.
>
> ---
>
> ### Responses to Weaknesses
>
> ---
>
> > 1. 3D style loss justification is empirical. The 3D voxel-statistics loss gives better LPIPS/RMSE consistency, but the paper does not analyze potential side effects such as global color bleeding or loss of local style detail in poorly observed regions. Some intuition or failure-case visualization would strengthen the argument that voxel-statistics matching is fundamentally better than 2D scene-level matching.
>
> Thanks for recognizing that our proposed voxel-level 3D style loss yields improved multi-view consistency, and for raising a valuable point regarding the motivation behind this design.
>
> First, our 3D style loss is view-independent because the style statistics are computed in the shared 3D voxel space, neither in the simple image plane nor in the 2D-based multi-view scene. Features from all views are first mapped to their corresponding 3D locations and fused in the voxel grid. Each voxel, therefore, aggregates appearance information from all valid observations, and the style loss is applied to these fused voxel features. As a result, the style supervision is anchored in 3D world coordinates, independent of the camera viewpoint, which avoids the view-dependent inconsistencies inherent in 2D image- or scene-level losses. For clarity, if we ignore the batch dimension and multi-level structure, the proposed 3D style loss is formalized in Algorithm 1.
>
> **Algorithm 1: 3D Voxel-based AdaIN Style Loss**
> **Inputs:**  Rendered features 𝓡, Style features 𝓢, 3D points 𝓟, Confidence 𝓒, Valid mask 𝓜
> **Output:**  3D style loss 𝓛
> ```pseudo
> Initialize 𝓛 ← 0
> Resize(𝓒, 𝓜, 𝓟) → match resolution of 𝓡
> # Fuse all views into a voxel grid
> G ← VoxelizeAndFuse({𝓡,v}_{v=1..V},{𝓟,v}_{v=1..V},{𝓒,v}_{v=1..V},{𝓜,v}_{v=1..V})
> # Compute BN statistics for two feature tensors
> (μ_g, σ_g) ← MeanStd(G), (μ_s, σ_s) ← MeanStd(𝓢)
> 𝓛 ← 𝓛 + ‖μ_g − μ_s‖² + ‖σ_g − σ_s‖²
> return 𝓛 = λ · 𝓛
> ```
>
> In addition to improved consistency, **3D loss also enhances the artistic score**, as shown in Table 1. Together, these results indicate that enforcing style consistency at the voxel level leads ot more coherent stylized textures across views.
>
> ***Table 1. Artistic comparisons of different style loss designs***
>
> | Style Loss | ArtScore↑ |
> |:-----------|:----------:|
> | Image loss | $5.96$ |
> | Scene loss | $9.54$ |
> | **3D loss** | $\mathbf{9.62}$ |
>
> Regarding potential side effects, we have carefully evaluated a wide range of CO3D scenes, e.g., skateboards, donuts, pizzas, (see **Page 17-19, Figure 8-10**) under 50 diverse test styles (see **Page 22-23, Figure 14–15**), and we have not observed global color bleeding or loss of local style details that can be attributed to the voxel-level 3D style loss. For example, **Page 17 Figure 8** shows that the 3D style loss better preserves the wooden floor textures while achieving faithful stylization. We will release our implementations of all three style losses to enable further investigation by the community. As shown in **Page 18 Figure 9**, the 3D style loss is also able to stylize a larger portion of the table surface compared with the image-level loss. In **Page 19 Figure 10**, the seams on the tablecloth are well preserved under the 3D style loss, while they become partially distorted when using the image- or scene-level losses.
>
> However, we have observed several failure cases where the overall ***Stylos*** is less effective. In particular, ***Stylos*** tends to perform noticeably worse on high-frequency, highly cluttered scenes, such as dense foliage and wire-like structures, and on style categories with strong global lighting shifts or extreme color palettes. In these cases, the model may struggle to maintain fine geometric detail or may over-saturate the stylized output. We have added multiple failure cases to the Appendix. More specifically, the failure cases can be grouped into three categories:
> * Failures caused by inaccurate scene reconstruction, such as missing geometric details or incorrect region colors, as shown in the revised paper **Page 20 Figure 11**;
> * Failures caused by generating colors that are absent from the original style images, as shown in the revised paper **Page 21 Figure 12**;
> * Failures caused by introducing blurry or over-smoothed style colors, as shown in the revised paper **Page 21 Figure 13**.
>
>
> We have revised both the main paper and the appendix to include more detailed explanations about our loss designs with additional visual examples, enabling readers to better understand the differences among the three style losses and limitations of ***Stylos***. We expect more future work on improving the current version of ***Stylos***.

---

> ### Author Response · Authors · 2025-11-25
> **Response to Reviewer LtqC, Part II**
>
> > 2. Scalability limits. Although Stylos is advertised as handling up to “hundreds of views,” the scaling experiment shows visible degradation beyond ~32 views per batch (edge artifacts, instability in the Gaussian representation). The method may still be quite useful, but this practical constraint, probably brought by VGGT, should be communicated more prominently.
>
> Thank you for highlighting this important point. We agree that the scalability limitations should be communicated more clearly. While ***Stylos*** is designed to support large numbers of input views, our scaling experiment indeed shows that performance begins to degrade when exceeding approximately 32 views per batch, with artifacts such as edge inconsistencies and reduced stability in the Gaussian representation. We agree that this constraint is primarily inherited from the underlying VGGT [1] backbone, whose attention and depth-prediction modules become less stable under very dense multi-view inputs. We have updated the paper to more prominently discuss this practical limitation. Importantly, despite this constraint, ***Stylos*** remains effective in typical multi-view reconstruction settings, and we view improving dense-view stability, either through backbone modifications or hierarchical batching strategies, as a promising direction for future work.
>
> ---
>
> ### Responses to Questions
>
> ---
>
> > 1. Failure cases of the 3D style loss. Are there any visual failure cases where voxelized style statistics hurt local detail (e.g., oversmoothing, texture bleeding in occluded regions)? Right now we only see success cases and quantitative gains.
>
> Please refer to our response to Weakness 1.
>
> ---
>
> #### Reference
> [1] Wang, Jianyuan, et al. "Vggt: Visual geometry grounded transformer." Proceedings of the Computer Vision and Pattern Recognition Conference. 2025.

---

> ### Comment · Reviewer_LtqC · 2025-11-27
>
> Thanks for the rebuttal. It resolves my concerns. I've read through all the other reviews, and I think this work deserves the rating of 8. I will keep my rating unchanged.

---

> > ### Author Response · Authors · 2025-11-27
> >
> > Thank you for taking the time to review our rebuttal and the other reviews. We are glad that our clarifications addressed the concerns you raised, and we appreciate your continued evaluation of our submission. Thank you again!

---

### Official Review · Reviewer_Sv2n · 2025-10-29

**Soundness:** 3
**Presentation:** 3
**Contribution:** 2
**Rating:** 6
**Confidence:** 4

**Summary:**

The paper introduces StylOS, an efficient framework for view-consistent 3D style transfer using 3D Gaussian Splatting (3DGS). The main technical contribution is a single-forward inference model, built upon VGGT and AnySplat, that directly predicts the stylized 3DGS representation. This approach enables optimization-free, geometry-aware synthesis from unposed content views (from a single image to a multi-view collection) and a reference style image. By bypassing time-consuming per-scene optimization, StylOS achieves fast processing and superior generalization to unseen scenes and styles.

**Strengths:**

1. **Single-Forward Efficiency**: The most compelling contribution is the ability to perform high-quality 3D style transfer in a single forward pass without per-scene optimization. This drastically reduces processing time and makes 3D stylization scalable for real-time or large-scale applications.

2. **Transformer Integration and Unposed Capability**: By utilizing a Transformer architecture (built upon VGGT) to directly predict the parameters of the stylized 3D Gaussian Splatting representation, the method bypasses the traditional optimization pipeline and simultaneously enables robust handling of unposed content (single or multi-view). This significantly lowers the barrier to entry for users, as complex camera pose estimation is not required.

3. **Generalization and Robustness**: The method demonstrates excellent generalization capabilities across unseen content scenes, object categories, and style images, suggesting robust learning of disentangled style and geometry features.

**Weaknesses:**

1. **Limited Technical Novelty in Overall Framework**: The overall conceptual framework of achieving fast, optimization-free 3D stylization by integrating a feed-forward 3D reconstruction model (VGGT/AnySplat) with cross-attention for style injection is highly similar to previous works like Styl3R (NeurIPS 2025, published on arXiv in May 2025), which uses Dust3R. While StylOS's integration of VGGT allows it to process a wider and more flexible range of input views, the core technical contribution of the idea of a single-forward 3D stylization framework is substantially weakened by this conceptual overlap. The authors should better address this overlap and clearly articulate how their specific implementation of the style injection and 3DGS prediction provides unique advantages over Styl3R's architecture.

2. **Style Fidelity vs. Optimization**: Although the speed advantage is clear, a deeper discussion or visual comparison is needed to evaluate the potential trade-off in style fidelity against state-of-the-art optimization-based 3D style transfer methods like 𝒢‐Style: Stylized Gaussian Splatting or ARF. Are there certain styles (e.g., fine-grained texture styles) where the feed-forward approach visibly struggles compared to optimization?

3. **Minor Formatting Concern (Font)**: It appears the authors may have modified the default font used in the ICLR style configuration.

**Questions:**

1. **Style Control and Blending**: Can the authors demonstrate fine-grained control over the stylization process? Specifically:
a. Can the system adjust the content-style trade-off weight post-inference to control the strength of the stylistic transfer?
b. Can the model perform multi-style blending by interpolating or averaging the style embeddings from two or more distinct style images?

2. **Input and Output Resolution**: What is the range of resolutions for the input content views (single or multi-view) and the output rendering that the model is designed to support? Are there inherent limitations or scaling constraints related to the Transformer architecture or 3DGS representation density?

---

> ### Author Response · Authors · 2025-11-25
> **Response to Reviewer Sv2n, Part I**
>
> We sincerely thank the reviewer for raising these reasonable concerns and for providing valuable suggestions on our work. We have made every effort to conduct comprehensive experiments to further strengthen the persuasiveness of our method, which we believe helps clarify the reviewer’s concerns. If there are any remaining questions, we would be more than happy to address them. Our responses are as follows.
>
> ---
>
> ### Response to Weaknesses
>
> ---
>
> 1. > Limited Technical Novelty in Overall Framework: The overall conceptual framework of achieving fast, optimization-free 3D stylization by integrating a feed-forward 3D reconstruction model (VGGT/AnySplat) with cross-attention for style injection is highly similar to previous works like Styl3R (NeurIPS 2025, published on arXiv in May 2025), which uses Dust3R. While StylOS's integration of VGGT allows it to process a wider and more flexible range of input views, the core technical contribution of the idea of a single-forward 3D stylization framework is substantially weakened by this conceptual overlap. The authors should better address this overlap and clearly articulate how their specific implementation of the style injection and 3DGS prediction provides unique advantages over Styl3R's architecture.
>
> Thank you for raising the point regarding *Styl3R* [1], which has already been cited in our initial submission (lines 99 and 314). At the time of our submission, no code or reproducible models were available, and our method was developed independently, without access to their implementation details or design choices. Beyond the differences in backbone choice, our research focus and technical contributions are different from *Styl3R* [1], which will be clarified as follows.
>
> (1) Our style injection mechanism follows the standard cross-attention formulation widely used in transformer-based 2D style transfer methods such as StyTr² [2], where content tokens serve as queries and style tokens serve as keys/values. Extending this to the 3D multi-view setting requires new design considerations regarding global attention, per-frame attention, and hybrid cross-attention. We build on the VGGT geometry aggregator, which employs hybrid cross-attention and empirically achieves better geometric stability. Our ablation study, as shown in the **following Table 1**, further shows that global-only cross-attention produces significantly better structural consistency than the other two strategies. *Styl3R* [1] does not discuss these distinctions or analyze style-injection mechanisms in detail.
>
> ***Table 1. Ablation on coupling strategies***
>
> Reconstruction quality measured by PSNR↑, SSIM↑, LPIPS↓ on CO3D.
>
> | Global | Frame | Skateboard | Skateboard | Skateboard | Pizza | Pizza | Pizza | Donut | Donut | Donut |
> |:------:|:-----:|:----------:|:----------:|:----------:|:-----:|:-----:|:------:|:-----:|:------:|:-------:|
> |        |       | PSNR↑ | SSIM↑ | LPIPS↓ | PSNR↑ | SSIM↑ | LPIPS↓ | PSNR↑ | SSIM↑ | LPIPS↓ |
> | ✓ | ✓ | $21.12$ | $0.6858$ | $0.2821$ | $19.78$ | $0.5939$ | $0.3326$ | $21.39$ | $0.7198$ | $0.3264$ |
> |   | ✓ | $20.93$ | $0.6917$ | $0.2912$ | $19.72$ | $0.5940$ | $0.3405$ | $21.40$ | $0.7167$ | $0.3340$ |
> | ✓ |   | $\mathbf{21.68}$ | $\mathbf{0.7043}$ | $\mathbf{0.2684}$ | $\mathbf{20.57}$ | $\mathbf{0.6177}$ | $\mathbf{0.3110}$ | $\mathbf{22.09}$ | $\mathbf{0.7362}$ | $\mathbf{0.3125}$ |
>
> (2) Style loss formulation. Directly applying the image-level style loss without any multi-view or 3D consistency enforcement leads to noticeable artifacts and multi-view inconsistency. To address this, we study two alternative formulations: a scene-level style loss and a voxel-level style loss. Our ablation experiments indicate that the voxel-level formulation provides more stable style propagation and better cross-view coherence, which motivates our final design choice. *Styl3R* reports “computing the difference in mean and variance between rendered novel-view images and the reference style image”, which is comparable to our image-level baseline style loss. To the best of our knowledge, *Styl3R* [1] may not enforce multi-view style consistency in their style loss design, which is essential for stable and coherent 3D stylization. In contrast, our proposed voxel-level 3D style loss well enforces multi-view stylization consistency as reported in the following Table 2, compared to the baseline image-level style loss, and the scene-level variant.
>
> ***Table 2. Consistency comparison of different style loss designs***
>
> | Style Loss | Short-range | Short-range | Long-range | Long-range |        |
> |:----------|:-----------:|:-----------:|:-----------:|:-----------:|:----------:|
> |           | LPIPS↓ | RMSE↓ | LPIPS↓ | RMSE↓ | ArtScore↑ |
> | Image loss | $0.046$ | $0.035$ | $0.147$ | $0.135$ | $5.96$ |
> | Scene loss | $0.048$ | $0.037$ | $0.147$ | $0.133$ | $9.54$ |
> | **3D loss** | $\mathbf{0.046}$ | $\mathbf{0.033}$ | $\mathbf{0.140}$ | $\mathbf{0.128}$ | $\mathbf{9.62}$ |

---

> ### Author Response · Authors · 2025-11-25
> **Response to Reviewer Sv2n, Part II**
>
> (3) Because ***Stylos*** is built on the VGGT backbone, our method supports training with a flexible and variable number of input views, rather than requiring a fixed number of images per scene. This flexibility helps the model better learn view-consistent style features and adapt to scenes with different coverage levels.
>
> We will clarify these distinctions and our independent development process in the revised manuscript.
>
> ---
>
> 2. > Style Fidelity vs. Optimization: Although the speed advantage is clear, a deeper discussion or visual comparison is needed to evaluate the potential trade-off in style fidelity against state-of-the-art optimization-based 3D style transfer methods like 𝒢‐Style: Stylized Gaussian Splatting or ARF. Are there certain styles (e.g., fine-grained texture styles) where the feed-forward approach visibly struggles compared to optimization?
>
> We thank the reviewer for recognizing our speed advantage as shown in the revised paper **Page 9 Table 4**. In terms of visual comparisons, we provide additional quantitative comparisons in the **following Table 3-6**. We emphasize that the 50 varying style images for evaluation were not used for training Stylos. As shown in these tables, our stylization method demonstrates consistently strong performance compared to the per-scene methods. The updated **Page 10 Figure 5** in the revised paper shows that, compared to the per-scene methods, ***Stylos*** demonstrates a weaker ability to preserve clear high-frequency details such as the wood texture on the truck trailer at the top row of Figure 5 with certain types of styles.
>
> More specifically, the failure cases of ***Stylos*** can be grouped into three categories:
> * Failures caused by inaccurate scene reconstruction, such as missing geometric details or incorrect region colors, as shown in the appendix **Page 20 Figure 11**;
> * Failures caused by generating colors that are absent from the original style images, as shown in the appendix **Page 21 Figure 12**;
> * Failures caused by introducing blurry or over-smoothed style colors, as shown in the appendix **Page 21 Figure 13**.
>
> ***Table 3. Short-range consistency.***
>
> | Method | Train | Train | Truck | Truck | M60 | M60 | Garden | Garden |
> |:------:|:-----:|:-----:|:-----:|:-----:|:---:|:----:|:-------:|:---------:|
> |        | LPIPS↓ | RMSE↓ | LPIPS↓ | RMSE↓ | LPIPS↓ | RMSE↓ | LPIPS↓ | RMSE↓ |
> | StyleGaussian [3] | $0.033$ | $0.038$ | $0.031$ | $0.034$ | $0.038$ | $0.037$ | $0.069$ | $0.061$ |
> | G-Style [4] | $0.042$ | $0.052$ | $0.032$ | $0.035$ | $0.038$ | $0.034$ | $0.066$ | $0.070$ |
> | StylizedGS [5] | $\mathbf{0.016}$ | $\mathbf{0.023}$ | $\mathbf{0.013}$ | $\mathbf{0.019}$ | $\mathbf{0.017}$ | $\mathbf{0.020}$ | $\underline{0.056}$ | $\mathbf{0.041}$ |
> | SGSST [6] | $0.038$ | $0.047$ | $0.039$ | $0.047$ | $0.044$ | $0.049$ | $0.084$ | $0.090$ |
> | Styl3R [1] | -- | -- | $0.061$ | $0.036$ | $0.066$ | $0.040$ | $0.105$ | $0.067$ |
> | ***Stylos*** (ours) | $\underline{0.030}$ | $\underline{0.026}$ | $\underline{0.028}$ | $\underline{0.021}$ | $\underline{0.035}$ | $\underline{0.024}$ | $\mathbf{0.047}$ | $\underline{0.044}$ |
>
> ***Table 4. Long-range consistency.***
>
> | Method | Train | Train | Truck | Truck | M60 | M60 | Garden | Garden |
> |:------:|:-----:|:-----:|:-----:|:-----:|:---:|:----:|:-------:|:---------:|
> |        | LPIPS↓ | RMSE↓ | LPIPS↓ | RMSE↓ | LPIPS↓ | RMSE↓ | LPIPS↓ | RMSE↓ |
> | StyleGaussian [3] | $0.067$ | $0.072$ | $0.086$ | $0.077$ | $0.091$ | $0.091$ | $0.177$ | $\underline{0.141}$ |
> | G-Style [4] | $0.098$ | $0.120$ | $0.095$ | $0.093$ | $0.104$ | $0.095$ | $0.180$ | $0.175$ |
> | StylizedGS [5] | $\mathbf{0.040}$ | $\mathbf{0.065}$ | $\mathbf{0.049}$ | $\mathbf{0.064}$ | $\mathbf{0.065}$ | $\underline{0.086}$ | $\mathbf{0.100}$ | $0.202$ |
> | SGSST [6] | $0.087$ | $0.108$ | $0.119$ | $0.120$ | $0.130$ | $0.128$ | $0.221$ | $0.222$ |
> | Styl3R [1] | -- | -- | $0.116$ | $0.100$ | $0.147$ | $0.143$ | $0.146$ | $0.145$ |
> | ***Stylos*** (ours) | $\underline{0.051}$ | $\underline{0.056}$ | $\underline{0.074}$ | $\underline{0.069}$ | $\underline{0.083}$ | $\mathbf{0.082}$ | $\underline{0.139}$ | $\mathbf{0.134}$ |

---

> ### Author Response · Authors · 2025-11-25
> **Response to Reviewer Sv2n, Part III**
>
> ***Table 5. Stylization quality (ArtScore / ArtFID) and Stylization Time.***
>
> | Method | Train | Train | Truck | Truck | M60 | M60 | Garden | Garden |
> |:------:|:-----:|:-----:|:-----:|:-----:|:---:|:----:|:-------:|:---------:|
> |        | Score↑ | FID↓ | Score↑ | FID↓ | Score↑ | FID↓ | Score↑ | FID↓ |
> | StyleGaussian [3] | $0.78$ | $52.79$ | $5.76$ |$44.93$ | $8.63$ | $47.48$ | $\mathbf{9.38}$ | $41.14$ |
> | G-Style [4] | **$\mathbf{9.52}$** | **$\mathbf{23.24}$** | **$\underline{9.67}$** | **$\mathbf{22.15}$** | **$\mathbf{9.73}$** | **$\mathbf{22.36}$** | **$8.98$** | **$\mathbf{25.76}$** |
> | StylizedGS [5] | $4.95$ | $40.79$ | $6.45$ | $42.07$ | $7.39$ | $53.24$ | $6.99$ | $65.15$ |
> | SGSST [6] | $1.84$ | $38.24$ | $5.34$ | $32.34$ | $5.26$ | $38.73$ | $4.89$ | $33.54$ |
> | Styl3R [1] | -- | -- | $2.94$ | $34.11$ | $2.96$ | $29.86$ | $4.09$ | $38.28$ |
> | ***Stylos*** (ours) | $\underline{9.50}$ | $\underline{26.40}$ | $\mathbf{9.70}$ | $\underline{28.71}$ | $\underline{9.37}$ | $\underline{27.44}$ | $\underline{9.34}$ | $\underline{28.06}$ |
>
> ***Table 6. Stylization Quality (CFSD & Color Matching Loss)***
>
> We follow ***StyleID*** [2] and additionally compare the CFSD and HistoGAN loss metrics.
> | Method | Train | Train | Truck | Truck | M60 | M60 | Garden | Garden |
> |:------:|:-----:|:-----:|:-----:|:-----:|:---:|:----:|:-------:|:--------:|
> |        | CFSD↓ | HistoGAN Loss↓ | CFSD↓ | HistoGAN Loss↓ | CFSD↓ | HistoGAN Loss↓ | CFSD↓ | HistoGAN Loss↓ |
> | StyleGaussian [3] | $\underline{0.190}$ | $0.418$ | $0.131$ | $0.382$ | $0.138$ | $0.467$ | $0.189$ | $0.480$ |
> | G-Style [4] | **$\mathbf{0.165}$** | **$\mathbf{0.185}$** | **$\underline{0.087}$** | **$\mathbf{0.176}$** | **$\mathbf{0.092}$** | **$\mathbf{0.208}$** | $0.103$ | **$\mathbf{0.179}$** |
> | StylizedGS [5] | $0.216$ | $0.396$ | $\mathbf{0.084}$ | $0.417$ | $0.102$ | $0.443$ | **$\mathbf{0.074}$** | $0.570$ |
> | SGSST [6] | $0.220$ | $0.257$ | $0.177$ | $\underline{0.196}$ | $0.204$ | $0.264$ | $0.233$ | $\underline{0.228}$ |
> | Styl3R [1] | $0.244$ | $0.364$ | $0.102$ | $\underline{0.350}$ | $0.124$ | $0.314$ | $0.097$ | $0.335$ |
> | ***Stylos*** (ours) | $0.223$ | $\underline{0.241}$ | $\mathbf{0.084}$ | $0.237$ | $\underline{0.098}$ | $\underline{0.252}$ | $\underline{0.080}$ | $0.242$ |
>
> ---
>
> 3. > Minor Formatting Concern (Font): It appears the authors may have modified the default font used in the ICLR style configuration.
>
> Thanks for bringing up this font style issue to our attention. It has been corrected in the revised submission. The problem was likely caused by the LaTeX compiler unintentionally overriding the default font settings.
>
> ---
>
> ### Response to Questions:
>
> ---
>
> 1. > Style Control and Blending: Can the authors demonstrate fine-grained control over the stylization process? Specifically: a. Can the system adjust the content-style trade-off weight post-inference to control the strength of the stylistic transfer? b. Can the model perform multi-style blending by interpolating or averaging the style embeddings from two or more distinct style images?
>
> We had not previously evaluated fine-grained style control, but we conducted additional experiments in response to the reviewer’s question (please see the visual results in the revised paper **Page 10, Figure 6**). First, we performed linear interpolation between the embeddings of two distinct style images, and we observed a smooth and coherent transition between the two styles, confirming that our model naturally supports multi-style blending. In addition, because the model is trained to reconstruct the original appearance when the “style image” matches the input content image, we conducted a complementary experiment interpolating between the reconstruction (content) embedding and a stylized embedding. This produces a continuous spectrum of outputs with varying degrees of stylization, demonstrating that our method can indeed provide post-inference control over the content–style trade-off. These results indicate that our approach enables both multi-style blending and adjustable stylization strength without requiring retraining or additional optimization.

---

> ### Author Response · Authors · 2025-11-25
> **Response to Reviewer Sv2n, Part IV**
>
> 2. > Input and Output Resolution: What is the range of resolutions for the input content views (single or multi-view) and the output rendering that the model is designed to support? Are there inherent limitations or scaling constraints related to the Transformer architecture or 3DGS representation density?
>
> The input resolution is constrained to 448 pixels on the longest side, with the aspect ratio randomized between 0.5 and 1.0. The stylized renderings are produced at the same resolution during training to ensure consistency between reconstruction and appearance transformation.
>
> Because our backbone uses VGGT, the main architectural limitation is the model’s inability to process very long sequences of input views in a single forward pass, as standard self-attention scales quadratically with the number of image tokens.
>
> For the 3DGS representation, the effective number of Gaussians is determined by the voxelization step, leading to a lower density than in pixel-wise Gaussian representations, which are usually used in a feedforward Gaussian model.
>
> ---
>
> #### Reference
>
> [1] Wang, Peng, Xiang Liu, and Peidong Liu. "Styl3R: Instant 3D Stylized Reconstruction for Arbitrary Scenes and Styles." arXiv preprint arXiv:2505.21060 (2025).
>
> [2] Chung, Jiwoo, Sangeek Hyun, and Jae-Pil Heo. "Style injection in diffusion: A training-free approach for adapting large-scale diffusion models for style transfer." Proceedings of the IEEE/CVF conference on computer vision and pattern recognition. 2024.
>
> [3] Liu, Kunhao, et al. "Stylegaussian: Instant 3d style transfer with gaussian splatting." SIGGRAPH Asia 2024 Technical Communications. 2024. 1-4.
>
> [4] Kovács, Áron Samuel, Pedro Hermosilla, and Renata G. Raidou. "G‐Style: Stylized Gaussian Splatting." Computer Graphics Forum. Vol. 43. No. 7. 2024.
>
> [5] Zhang, Dingxi, et al. "Stylizedgs: Controllable stylization for 3d gaussian splatting." arXiv preprint arXiv:2404.05220 (2024).
>
> [6] Galerne, Bruno, et al. "SGSST: Scaling Gaussian Splatting Style Transfer." Proceedings of the Computer Vision and Pattern Recognition Conference. 2025.

---

> > ### Comment · Reviewer_Sv2n · 2025-11-26
> >
> > I thank the authors for their detailed response. The clarification on the difference from​ Styl3R and the analysis of​ failure cases and comparisons with​ the optimization-based method are informative. Thus, I have decided to maintain my positive rating.

---

> > > ### Author Response · Authors · 2025-11-27
> > >
> > > Thank you for your feedback and for reconsidering our clarifications on the differences from *Styl3R* [1], the failure-case analysis, and the comparison with optimization-based approaches. We appreciate your time in reviewing both our submission and rebuttal, as well as your continued positive evaluation. Thank you again.

---

### Official Review · Reviewer_Y42i · 2025-10-30

**Soundness:** 4
**Presentation:** 4
**Contribution:** 4
**Rating:** 8
**Confidence:** 5

**Summary:**

The paper introduces a feedforward stylized Gaussian method to achieve efficient, high-quality, and visually pleasing 3D scene reconstruction and stylization in a single step of model inference. The proposed approach is built upon VGGT, taking multi-view scene images and a style exemplar image as input. The model predicts the depth and camera poses for each view, while the predicted pixel Gaussian attributes are merged into voxels using confidence-based weights following the AnySplat framework. The results, as demonstrated in the figures and supplementary video, are remarkable.

This feedforward stylized Gaussian model has the potential to significantly impact the field. Artistic stylization, a cornerstone of human creativity throughout history, plays a vital role in bridging the technical and emotional aspects of visual representation. In the context of 3D world reconstruction, artistic stylization becomes even more critical, as it encapsulates the essence of reality through the lens of human emotion, emphasizing mood, character, and individuality. By enabling stylized 3D scene reconstruction without the need for test-time training, this model addresses a significant challenge and represents a major step forward in both technical and artistic innovation.

**Strengths:**

The technical strengths of the paper can be summarized as follows:

1. This work is the first to achieve a stylized 3D world in a fully feedforward manner. From the early days of this field, StylizedNeRF introduced the concept of reconstructing 3D scenes using NeRF, accompanied by a conditional stylization module. While it generalized across styles, it required per-scene optimization. ARF, on the other hand, achieved superior results using a style-matching loss and a deferred backward strategy, but it was not generalized across scenes or styles. Both methods, being NeRF-based, faced limitations due to the high GPU demands during training, requiring solutions like mutual learning and deferred optimization to mitigate these challenges. Similarly, StyleRF was also restricted to per-scene optimization.

   In the era of 3DGS, StylizedGS achieved impressive results in 3D stylization but remained constrained by per-scene fitting. This paper, however, marks a significant milestone by creating a stylized 3D world using a tuned foundation model like VGGT, combined with a generalized stylization module, enabling a fully feedforward pipeline. This eliminates the need for lengthy optimization processes. There is no more waiting hours to transform multi-view images into an artistic 3D world. This breakthrough paves the way for future research to focus on achieving even faster and higher-quality stylizations using similar feedforward approaches.

2. The results presented in the paper are both impressive and inspiring. The examples, particularly in the supplementary video, showcase how realistic 3D scenes can be transformed into artistic styles such as cartoon, sketch, and painting. The stylization maintains consistency across views without any flickering artifacts, which is a significant achievement. Moreover, obtaining such high-quality results in a single feedforward inference is truly remarkable and highlights the potential for real-time applications in this domain.

**Weaknesses:**

1. The most significant contribution of this work is the introduction of a feedforward 3D stylization model, whose core advantages lie in faster inference and reduced memory usage. However, these critical metrics are not evaluated in the experiments. The paper should include a comparison of stylization time across prior works, from the earliest StylizedNeRF, ARF, and StyleRF to the more recent StyleGaussian and StylizedGaussian. For methods requiring per-scene fitting, the stylization time should account for both training and rendering times. Such comparisons would provide readers with a clearer perspective on how this work advances the field in terms of efficiency and practicality.

2. The comparison between NeRF-based and Gaussian-based stylization approaches is not adequately addressed. While StyleRF focuses on zero-shot stylization, its stylization quality is not the strongest. A detailed comparison with stylization examples from StylizedNeRF and similar NeRF-based methods is essential to highlight the advantages of the proposed approach. Moreover, StylizedGS, which outperforms StyleGS, should be included as the representative Gaussian-based stylization method in the comparisons. Incorporating these comparisons would strengthen the paper by demonstrating the superiority of the proposed method over both NeRF-based and Gaussian-based approaches.

**Questions:**

1. **Include Inference Speed Comparisons**:
   Adding a comparison of inference speed would highlight the advantages of the proposed method and make the paper more compelling. Demonstrating faster processing times would provide a stronger justification for the model's practical contributions to the field.

2. **Compare with StylizedNeRF and StylizedGaussian**:
   Include comparisons with StylizedNeRF and StylizedGaussian, as their stylization results are superior to those of StyleRF and StyleGaussian, respectively. The latter two methods primarily focus on zero-shot and fast stylization but rely on a CNN decoder, which means they are not strictly stylized **radiance** fields but rather stylization **feature** fields.

---

> ### Author Response · Authors · 2025-11-25
> **Response to Reviewer Y42i, Part I**
>
> We appreciate that the reviewer recognizes the impressive stylization results achieved by ***Stylos***, as presented in our video demo, which has been recently updated in the revised supplementary material. We also thank the reviewer for raising two valuable points: (1) stylization time comparisons, and (2) comparisons with non-zero-shot per-scene training methods. We have carefully reproduced the results of recent 3D stylization methods on the Tanks and Temples dataset, and we summarize our findings and responses as follows.
>
> ---
>
> ### Response to Weaknesses and Questions
>
> ---
>
> > 1. Include Inference Speed Comparisons:
> Adding a comparison of inference speed would highlight the advantages of the proposed method and make the paper more compelling. Demonstrating faster processing times would provide a stronger justification for the model's practical contributions to the field.
>
> Thank you for your thoughtful feedback and recognition of our work. We have included a comparison of stylization times across *StyleGaussian* [1], *G-Style* [2], *StylizedGS* [3], *SGSST* [4], and ***Stylos*** (ours). According to the following **Table 1**, the proposed ***Stylos*** achieves practical real-time 3D stylization. We have updated **Page 9 Table 4** with stylization times.
>
> ***Table 1: Stylization Time Comparison***
>
> All models are retrained and evaluated on a single NVIDIA GH200 GPU with the same hardware configuration. We follow the released codes and strictly follow the authors' training instructions and evaluation pipelines.
>
> | Methods | StyleGaussian [1] | G-Style [2] | StylizedGS [3] | SGSST [4] | ***Stylos*** (ours) |
> |:--------------:|:-----------------:|:----------------:|:---------------:|:----------:|:--------------------:|
> | Time↓          | $165 \,\mathrm{mins}$ | **$14.5 \,\mathrm{mins}$** | $13.3 \,\mathrm{mins}$ | $35.0 \,\mathrm{mins}$ | $\mathbf{0.05 \,\mathrm{secs}}$ |
> ---
>
> > 2. Compare with StylizedNeRF and StylizedGaussian:
> Include comparisons with StylizedNeRF and StylizedGaussian, as their stylization results are superior to those of StyleRF and StyleGaussian, respectively. The latter two methods primarily focus on zero-shot and fast stylization but rely on a CNN decoder, which means they are not strictly stylized radiance fields but rather stylization feature fields.
>
> Due to hardware constraints, we are unable to run *StylizedNeRF* [5] with its required package dependencies. Instead, we made our best effort to evaluate recent per-scene fitting models and report their results in the following **Table 2-5**.
>
> ***Table 2. Short-range consistency.***
>
> | Method | Train | Train | Truck | Truck | M60 | M60 | Garden | Garden |
> |:------:|:-----:|:-----:|:-----:|:-----:|:---:|:----:|:-------:|:---------:|
> |        | LPIPS↓ | RMSE↓ | LPIPS↓ | RMSE↓ | LPIPS↓ | RMSE↓ | LPIPS↓ | RMSE↓ |
> | StyleGaussian [1] | $0.033$ | $0.038$ | $0.031$ | $0.034$ | $0.038$ | $0.037$ | $0.069$ | $0.061$ |
> | G-Style [2] | $0.042$ | $0.052$ | $0.032$ | $0.035$ | $0.038$ | $0.034$ | $0.066$ | $0.070$ |
> | StylizedGS [3] | $\mathbf{0.016}$ | $\mathbf{0.023}$ | $\mathbf{0.013}$ | $\mathbf{0.019}$ | $\mathbf{0.017}$ | $\mathbf{0.020}$ | $\underline{0.056}$ | $\mathbf{0.041}$ |
> | SGSST [4] | $0.038$ | $0.047$ | $0.039$ | $0.047$ | $0.044$ | $0.049$ | $0.084$ | $0.090$ |
> | ***Stylos (ours)*** | $\underline{0.030}$ | $\underline{0.026}$ | $\underline{0.028}$ | $\underline{0.021}$ | $\underline{0.035}$ | $\underline{0.024}$ | $\mathbf{0.047}$ | $\underline{0.044}$ |
>
> ***Table 3. Long-range consistency.***
>
> | Method | Train | Train | Truck | Truck | M60 | M60 | Garden | Garden |
> |:------:|:-----:|:-----:|:-----:|:-----:|:---:|:----:|:-------:|:---------:|
> |        | LPIPS↓ | RMSE↓ | LPIPS↓ | RMSE↓ | LPIPS↓ | RMSE↓ | LPIPS↓ | RMSE↓ |
> | StyleGaussian [1] | $0.067$ | $0.072$ | $0.086$ | $0.077$ | $0.091$ | $0.091$ | $0.177$ | $\underline{0.141}$ |
> | G-Style [2] | $0.098$ | $0.120$ | $0.095$ | $0.093$ | $0.104$ | $0.095$ | $0.180$ | $0.175$ |
> | StylizedGS [3] | $\mathbf{0.040}$ | $\mathbf{0.065}$ | $\mathbf{0.049}$ | $\mathbf{0.064}$ | $\mathbf{0.065}$ | $\underline{0.086}$ | $\mathbf{0.100}$ | $0.202$ |
> | SGSST [4] | $0.087$ | $0.108$ | $0.119$ | $0.120$ | $0.130$ | $0.128$ | $0.221$ | $0.222$ |
> | ***Stylos (ours)*** | $\underline{0.051}$ | $\underline{0.056}$ | $\underline{0.074}$ | $\underline{0.069}$ | $\underline{0.083}$ | $\mathbf{0.082}$ | $\underline{0.139}$ | $\mathbf{0.134}$ |

---

> ### Author Response · Authors · 2025-11-25
> **Response to Reviewer Y42i, Part II**
>
> ***Table 4. Stylization Quality (ArtScore / ArtFID).***
>
> | Method | Train | Train | Truck | Truck | M60 | M60 | Garden | Garden |
> |:------:|:-----:|:-----:|:-----:|:-----:|:---:|:----:|:-------:|:---------:|
> |        | ArtScore↑ | ArtFID↓ | ArtScore↑ | ArtFID↓ | ArtScore↑ | ArtFID↓ | ArtScore↑ | ArtFID↓ |
> | StyleGaussian [1] | $0.78$ | $52.79$ | $5.76$ |$44.93$ | $8.63$ | $47.48$ | $\mathbf{9.38}$ | $41.14$ |
> | G-Style [2] | **$\mathbf{9.52}$** | **$\mathbf{23.24}$** | **$\underline{9.67}$** | **$\mathbf{22.15}$** | **$\mathbf{9.73}$** | **$\mathbf{22.36}$** | **$8.98$** | **$\mathbf{25.76}$** |
> | StylizedGS [3] | $4.95$ | $40.79$ | $6.45$ | $42.07$ | $7.39$ | $53.24$ | $6.99$ | $65.15$ |
> | SGSST [4] | $1.84$ | $38.24$ | $5.34$ | $32.34$ | $5.26$ | $38.73$ | $4.89$ | $33.54$ |
> | ***Stylos (ours)*** | $\underline{9.50}$ | $\underline{26.40}$ | $\mathbf{9.70}$ | $\underline{28.71}$ | $\underline{9.37}$ | $\underline{27.44}$ | $\underline{9.34}$ | $\underline{28.06}$ |
>
> ***Table 5. Stylization Quality (CFSD & Color Matching Loss)***
>
> We follow ***StyleID*** [6] and additionally compare the CFSD and HistoGAN loss metrics.
> | Method | Train | Train | Truck | Truck | M60 | M60 | Garden | Garden |
> |:------:|:-----:|:-----:|:-----:|:-----:|:---:|:----:|:-------:|:--------:|
> |        | CFSD↓ | HistoGAN Loss↓ | CFSD↓ | HistoGAN Loss↓ | CFSD↓ | HistoGAN Loss↓ | CFSD↓ | HistoGAN Loss↓ |
> | StyleGaussian [1] | $\underline{0.190}$ | $0.418$ | $0.131$ | $0.382$ | $0.138$ | $0.467$ | $0.189$ | $0.480$ |
> | G-Style [2] | **$\mathbf{0.165}$** | **$\mathbf{0.185}$** | **$\underline{0.087}$** | **$\mathbf{0.176}$** | **$\mathbf{0.092}$** | **$\mathbf{0.208}$** | $0.103$ | **$\mathbf{0.179}$** |
> | StylizedGS [3] | $0.216$ | $0.396$ | $\mathbf{0.084}$ | $0.417$ | $0.102$ | $0.443$ | **$\mathbf{0.074}$** | $0.570$ |
> | SGSST [4] | $0.220$ | $0.257$ | $0.177$ | $\underline{0.196}$ | $0.204$ | $0.264$ | $0.233$ | $\underline{0.228}$ |
> | ***Stylos (ours)*** | $0.223$ | $\underline{0.241}$ | $\mathbf{0.084}$ | $0.237$ | $\underline{0.098}$ | $\underline{0.252}$ | $\underline{0.080}$ | $0.242$ |
>
> As shown in the above tables, ***Stylos*** achieves consistently strong performance across both consistency and stylization metrics compared to recent per-scene fitting methods [1,2,3,4]. We have added these tables to the main manuscript and the appendix for a persuasive and comprehensive comparison with state-of-the-art 3D stylization approaches.
>
> ---
>
> #### Reference
>
> [1] Liu, Kunhao, et al. "Stylegaussian: Instant 3d style transfer with gaussian splatting." SIGGRAPH Asia 2024 Technical Communications. 2024. 1-4.
>
> [2] Kovács, Áron Samuel, Pedro Hermosilla, and Renata G. Raidou. "G‐Style: Stylized Gaussian Splatting." Computer Graphics Forum. Vol. 43. No. 7. 2024.
>
> [3] Zhang, Dingxi, et al. "Stylizedgs: Controllable stylization for 3d gaussian splatting." arXiv preprint arXiv:2404.05220 (2024).
>
> [4] Galerne, Bruno, et al. "SGSST: Scaling Gaussian Splatting Style Transfer." Proceedings of the Computer Vision and Pattern Recognition Conference. 2025.
>
> [5] Huang, Yi-Hua, et al. "Stylizednerf: consistent 3d scene stylization as stylized nerf via 2d-3d mutual learning." Proceedings of the IEEE/CVF conference on computer vision and pattern recognition. 2022.
>
> [6] Chung, Jiwoo, Sangeek Hyun, and Jae-Pil Heo. "Style injection in diffusion: A training-free approach for adapting large-scale diffusion models for style transfer." Proceedings of the IEEE/CVF conference on computer vision and pattern recognition. 2024.

---

### Official Review · Reviewer_YvUR · 2025-11-01

**Soundness:** 3
**Presentation:** 3
**Contribution:** 3
**Rating:** 6
**Confidence:** 3

**Summary:**

This paper presents StylOS, a single-forward 3D style transfer framework based on 3D Gaussian Splatting. The method employs a Transformer backbone with a dual-pathway design that separates geometry prediction from style injection: the geometry pathway maintains self-attention mechanisms to preserve geometric fidelity, while style is injected via global cross-attention to ensure multi-view consistency.

**Strengths:**

1. Proposes the first single-forward-pass 3D style transfer framework that requires no camera pre-calibration, offering significant practical value.

2. The method demonstrates strong scalability, supporting processing from a single view to hundreds of views.

**Weaknesses:**

1. Lacks fair comparison with per-scene optimization methods under the same computational budget.

2. Critical implementation details are missing regarding the voxelization operation: What is the voxel resolution? How are occlusions handled? What is the specific implementation of confidence weighting? Do these factors have significant impact on the results?

3. The weight parameters for each loss term are not provided.

4. The specific architecture of the Gaussian Adapter is not described in detail.

5. There is insufficient discussion on which scene types or style categories the method performs poorly on.

6. Why does global cross-attention not only improve style consistency but also enhance geometric fidelity? This seems counterintuitive and requires deeper analysis.

**Questions:**

See Weaknesses.

---

> ### Author Response · Authors · 2025-11-25
> **Response to Reviewer YvUR, Part I**
>
> We would like to sincerely thank the reviewer for insightful comments on our work. We take every comment seriously and hope our response can address the reviewer’s concerns. If there are any remaining questions, we are more than happy to address them. Our responses are as follows.
>
> ---
>
> ### Response to Weaknesses and Questions
>
> ---
>
> > 1. Lacks fair comparison with per-scene optimization methods under the same computational budget.
>
> We appricate the reviewer's suggestion. To ensure fairness, we follow the official training and testing pipelines of recent per-scene optimization methods (e.g., *G-Style* [2], *StylizedGS* [3], *SGSST* [4]) and a zero-shot method *StyleGaussian* [1].  In addition, we provide a direct comparison of stylization time. As shown in the **following Table 1-3**, ***Stylos*** ranks first or second across both consistency scores and artistic metrics. Moreover, ***Stylos*** achieves practical real-time stylization time.
>
> ***Table 1. Short-range consistency.***
>
> | Method | Train | Train | Truck | Truck | M60 | M60 | Garden | Garden |
> |:------:|:-----:|:-----:|:-----:|:-----:|:---:|:----:|:-------:|:---------:|
> |        | LPIPS↓ | RMSE↓ | LPIPS↓ | RMSE↓ | LPIPS↓ | RMSE↓ | LPIPS↓ | RMSE↓ |
> | StyleGaussian [1] | $0.033$ | $0.038$ | $0.031$ | $0.034$ | $0.038$ | $0.037$ | $0.069$ | $0.061$ |
> | G-Style [2] | $0.042$ | $0.052$ | $0.032$ | $0.035$ | $0.038$ | $0.034$ | $0.066$ | $0.070$ |
> | StylizedGS [3] | $\mathbf{0.016}$ | $\mathbf{0.023}$ | $\mathbf{0.013}$ | $\mathbf{0.019}$ | $\mathbf{0.017}$ | $\mathbf{0.020}$ | $\underline{0.056}$ | $\mathbf{0.041}$ |
> | SGSST [4] | $0.038$ | $0.047$ | $0.039$ | $0.047$ | $0.044$ | $0.049$ | $0.084$ | $0.090$ |
> | ***Stylos*** (ours) | $\underline{0.030}$ | $\underline{0.026}$ | $\underline{0.028}$ | $\underline{0.021}$ | $\underline{0.035}$ | $\underline{0.024}$ | $\mathbf{0.047}$ | $\underline{0.044}$ |
>
> ***Table 2. Long-range consistency.***
>
> | Method | Train | Train | Truck | Truck | M60 | M60 | Garden | Garden |
> |:------:|:-----:|:-----:|:-----:|:-----:|:---:|:----:|:-------:|:---------:|
> |        | LPIPS↓ | RMSE↓ | LPIPS↓ | RMSE↓ | LPIPS↓ | RMSE↓ | LPIPS↓ | RMSE↓ |
> | StyleGaussian [1] | $0.067$ | $0.072$ | $0.086$ | $0.077$ | $0.091$ | $0.091$ | $0.177$ | $\underline{0.141}$ |
> | G-Style [2] | $0.098$ | $0.120$ | $0.095$ | $0.093$ | $0.104$ | $0.095$ | $0.180$ | $0.175$ |
> | StylizedGS [3] | $\mathbf{0.040}$ | $\mathbf{0.065}$ | $\mathbf{0.049}$ | $\mathbf{0.064}$ | $\mathbf{0.065}$ | $\underline{0.086}$ | $\mathbf{0.100}$ | $0.202$ |
> | SGSST [4] | $0.087$ | $0.108$ | $0.119$ | $0.120$ | $0.130$ | $0.128$ | $0.221$ | $0.222$ |
> | ***Stylos*** (ours) | $\underline{0.051}$ | $\underline{0.056}$ | $\underline{0.074}$ | $\underline{0.069}$ | $\underline{0.083}$ | $\mathbf{0.082}$ | $\underline{0.139}$ | $\mathbf{0.134}$ |
>
> ***Table 3. Stylization quality (ArtScore / ArtFID) and Stylization Time.***
>
> For per-scene fitting required methods, stylization time accounts for both training and rendering times, which are marked with $^*$. More stylization metric results (such as CFSD, HigtoGAN loss, and etc.) are reported in Appendix Page 16 Table 5-6.
>
> | Method | Train | Train | Truck | Truck | M60 | M60 | Garden | Garden | Stylization |
> |:------:|:-----:|:-----:|:-----:|:-----:|:---:|:----:|:-------:|:---------:|:---------:|
> |        | Score↑ | FID↓ | Score↑ | FID↓ | Score↑ | FID↓ | Score↑ | FID↓ | Time↓ |
> | StyleGaussian [1] | $0.78$ | $52.79$ | $5.76$ |$44.93$ | $8.63$ | $47.48$ | $\mathbf{9.38}$ | $41.14$ | $165 \mathrm{mins} ^*$|
> | G-Style [2] | **$\mathbf{9.52}$** | **$\mathbf{23.24}$** | **$\underline{9.67}$** | **$\mathbf{22.15}$** | **$\mathbf{9.73}$** | **$\mathbf{22.36}$** | **$8.98$** | **$\mathbf{25.76}$** | $14.5 \mathrm{mins} ^*$ |
> | StylizedGS [3] | $4.95$ | $40.79$ | $6.45$ | $42.07$ | $7.39$ | $53.24$ | $6.99$ | $65.15$ | $13.3 \mathrm{mins} ^*$ |
> | SGSST [4] | $1.84$ | $38.24$ | $5.34$ | $32.34$ | $5.26$ | $38.73$ | $4.89$ | $33.54$ | $35.0 \mathrm{mins} ^*$ |
> | ***Stylos*** (ours) | $\underline{9.50}$ | $\underline{26.40}$ | $\mathbf{9.70}$ | $\underline{28.71}$ | $\underline{9.37}$ | $\underline{27.44}$ | $\underline{9.34}$ | $\underline{28.06}$ | $\mathbf{0.05} \mathrm{secs}$ |

---

> ### Author Response · Authors · 2025-11-25
> **Response to Reviewer YvUR, Part II**
>
> > 2. Critical implementation details are missing regarding the voxelization operation: What is the voxel resolution? How are occlusions handled? What is the specific implementation of confidence weighting? Do these factors have a significant impact on the results?
>
> Thank you for carefully reviewing our paper and bringing up these questions. We have added relevant implementation details to the paper. Specifically, the voxel size is set to 0.002. We do not explicitly model occlusions during voxelization; instead, occlusion effects are handled later in the pipeline through differentiable Gaussian rasterization ($\alpha$-compositing and depth ordering). Within each voxel, the Gaussian head predicts a per-Gaussian confidence value, which is converted into intra-voxel weights using a softmax. These weights are then used to compute confidence-weighted averages of all Gaussian attributes. This differentiable soft assignment stabilizes the clustering process and helps reduce redundancy among Gaussians. Finally, voxelization has a marginal effect on reconstruction accuracy but significantly improves scalability by slowing Gaussian count growth and reducing memory usage.
>
> ---
>
> > 3. The weight parameters for each loss term are not provided.
>
> Thank you for pointing this out. We have added the complete set of loss weights used in our experiments for clarity. Specifically, we use a style weight of 1.0, a content weight of 0.1, a CLIP consistency weight of 1.0, and a total variance regularization weight of 10.0.
>
> ---
>
> > 4. The specific architecture of the Gaussian Adapter is not described in detail.
>
> Thank you for highlighting the lack of architectural detail for the Gaussian Adapter. We have added a description of this adapter to the appendix. Specifically, the adapter splits the raw Gaussian parameters from the Gaussian head into scale, rotation, and spherical-harmonics components; applies softplus-based scale regularization and quaternion normalization; and constructs full 3D covariance matrices.
>
> ---
>
> > 5. There is insufficient discussion on which scene types or style categories the method performs poorly on.
>
> Thank you for raising this point. We have expanded our discussion to clarify the scenarios where the method is less effective. In particular, ***Stylos*** tends to perform noticeably worse on high-frequency, highly cluttered scenes (e.g., dense foliage, wire-like structures) and on style categories with strong global lighting shifts or extreme color palettes. In these cases, the model may struggle to maintain fine geometric detail or may over-saturate the stylized output. We have added multiple failure cases to the Appendix. More specifically, the failure cases can be grouped into three categories:
> * Failures caused by inaccurate scene reconstruction, such as missing geometric details or incorrect region colors, as shown in the **Page 20 Figure 11**;
> * Failures caused by generating colors that are absent from the original style images, as shown in the **Page 21 Figure 12**;
> * Failures caused by introducing blurry or over-smoothed style colors, as shown in the **Page 21 Figure 13**.

---

> ### Author Response · Authors · 2025-11-25
> **Response to Reviewer YvUR, Part III**
>
> > 6. Why does global cross-attention not only improve style consistency but also enhance geometric fidelity? This seems counterintuitive and requires deeper analysis.
>
> Thanks for pointing out this valuable point. To answer the question, let us review the architecture and training strategy of ***Stylos***. (1) Basically, ***Stylos*** consists of a style pathway which contributes to the color information for the predicted 3DGS representation, and a content pathway for the other 3DGS parameters. In the style pathway, we introduce a style aggregator with cross-attention to entangle the content and style tokens. (2) Our training pipeline consists of two stages. Stage 1 focuses on 3D scene reconstruction, where a random content view is used as the style reference; Stage 2 performs actual stylization using a real artistic style image.
>
> When designing  ***Stylos***, it is therefore essential to verify that introducing content-style cross-attention in the style aggregator does not harm the image quality of the rendered multi-view rgbs during Stage 1. Similar discussions about the importance of appearance-geometry entanglement in 3DGS and the potential effect of appearance to the final rendered outputs could be found in previous work such as ReGS [5], Texture-GS [6] and multiple 3DGS-based works.
>
> To this end, we compare three content–style coupling mechanisms (global, frame, and both) by calculating PSNR, SSIM and LPIPS between the rendered multi-view rgbs and their corresponding ground-thruth after Stage 1 training, using the first content view as the style reference. Interestingly, as shown in the **following Table 4**, the global-only variant achieves the best reconstruction accuracy and outperforms the hybrid method. This behavior is expected, because in frame attention, each view attends to the style tokens independently without interacting with the other views. Treating a single view as the style reference for all views forces each view to match its patch-level representation to the same reference individually, which distorts the rendered images. For example, in the **Page 7 Figure 2**, both the frame-only and the hybrid attention significantly blurs the rendered pizza box or the toppings.
>
> We have revised Sec. 3.2.2 to clarify content-style coupling methods, and added additional ablation analysis in the Appendix A.3.
>
> ***Table 4. Ablation on coupling strategies.***
>
> Reconstruction quality measured by PSNR↑, SSIM↑, LPIPS↓ on the CO3D test scenes.
>
> | Global | Frame | Skateboard | Skateboard | Skateboard | Pizza | Pizza | Pizza | Donut | Donut | Donut |
> |:------:|:-----:|:----------:|:----------:|:----------:|:-----:|:-----:|:------:|:-----:|:------:|:-------:|
> |        |       | PSNR↑ | SSIM↑ | LPIPS↓ | PSNR↑ | SSIM↑ | LPIPS↓ | PSNR↑ | SSIM↑ | LPIPS↓ |
> | ✓ | ✓ | $21.12$ | $0.6858$ | $0.2821$ | $19.78$ | $0.5939$ | $0.3326$ | $21.39$ | $0.7198$ | $0.3264$ |
> |   | ✓ | $20.93$ | $0.6917$ | $0.2912$ | $19.72$ | $0.5940$ | $0.3405$ | $21.40$ | $0.7167$ | $0.3340$ |
> | ✓ |   | $\mathbf{21.68}$ | $\mathbf{0.7043}$ | $\mathbf{0.2684}$ | $\mathbf{20.57}$ | $\mathbf{0.6177}$ | $\mathbf{0.3110}$ | $\mathbf{22.09}$ | $\mathbf{0.7362}$ | $\mathbf{0.3125}$ |
>
> ---
>
> #### Reference
>
> [1] Liu, Kunhao, et al. "Stylegaussian: Instant 3d style transfer with gaussian splatting." SIGGRAPH Asia 2024 Technical Communications. 2024. 1-4.
>
> [2] Kovács, Áron Samuel, Pedro Hermosilla, and Renata G. Raidou. "G‐Style: Stylized Gaussian Splatting." Computer Graphics Forum. Vol. 43. No. 7. 2024.
>
> [3] Zhang, Dingxi, et al. "Stylizedgs: Controllable stylization for 3d gaussian splatting." arXiv preprint arXiv:2404.05220 (2024).
>
> [4] Galerne, Bruno, et al. "SGSST: Scaling Gaussian Splatting Style Transfer." Proceedings of the Computer Vision and Pattern Recognition Conference. 2025.
>
> [5] Mei, Yiqun, Jiacong Xu, and Vishal Patel. "ReGS: Reference-based controllable scene stylization with gaussian splatting." Advances in Neural Information Processing Systems 37 (2024): 4035-4061.
>
> [6] Xu, Tian-Xing, et al. "Texture-gs: Disentangling the geometry and texture for 3d gaussian splatting editing." European Conference on Computer Vision. Cham: Springer Nature Switzerland, 2024.

---

### Author Response · Authors · 2025-11-25
**General Responses to All Reviewers**

We sincerely invite all readers, reviewers, and ACs to view our **updated demo video in the supplementary material**, which shows how the varying 3D scenes are stylized with different artistic images. The main contributions of ***Stylos*** are as follows.
1. We investigate different content–style coupling mechanisms, frame-only, global-only, and hybrid cross-attention, and analyze their effects on reconstruction accuracy. Our ablation studies in **Page 7 Table 1 and Figure 2** show that global-only cross-attention achieves the best scene reconstruction accuracy.
2. We start from the image-level style loss, which aligns statistics (mean and variance) per view independently, to the scene-level style loss, which aligns statistics considering all input views per scene, and finally the voxel-level 3d style loss, which aligns statistics in the aggregated voxel space. Our quantitative comparisons (**Page 7 Table 2**) and qualitative comparisons (**Page 8 Figure 3**, **Page 17-19 Figure 8-10**) clearly demonstrate the advantages of the 3D style loss over the other variants.
3. The proposed single-forward 3D stylization method, ***Stylos***, achieves practical real-time 3D stylization with consistently strong metrics scores as reported in **Page 9 Table 3-4**, **Page 16 Table 5-6**. Visual results are presented in the **Page 10 Figure 5**, **Page 24-33 Figure 16-25**, and the **updated demo video in the supplementary material**.

---

We thank all reviewers for their detailed and insightful comments. We carefully considered each comment and hope our responses adequately address the raised concerns. The reviewers' common concerns are summarized as follows.

* Failure cases of ***Stylos***. Briefly, the failure cases of our model can be categorized into three cases, (i) failures caused by inaccurate scene reconstruction, such as missing geometric details or incorrect region colors, as shown in the **Page 20 Figure 11**; (ii) failures caused by generating colors that are absent from the original style images, as shown in the **Page 21 Figure 12**; (iii) failures caused by introducing blurry or over-smoothed style colors, as shown in the **Page 21 Figure 13**.

* Lack Comparisons with per-scene optimization methods. Compared to recent per-scene optimization methods, ***Stylos*** demonstrates impressive stylization performance as rated by consistency scores, artistic scores, and stylization times, as shown in the **Page 9 Table 3-4** and **Page 16 Table 5-6**.

---
Inspired by the reviewers' valuable points, we have revised both the paper and the appendix. In addition to the changes mentioned above, the following modifications are summarized.
* We have added ArtScore comparisons into the **Page 7 Table 2** to further show the advantages of the 3D loss over the image and scene losses.
* We have additionally evaluated *G-Style* [1], *StylizedGS* [2], *SGSST* [3], and *Styl3R* [4] to demonstrate the comparable or even superior stylization performance of our method, while delivering faster stylization speed. We have also included a new metric, ArtFID [6], in the **Page 9 Table 4**. We have updated **Page 10 Figure 5** for visual comparisons with the aforementioned 3D stylization models. Moreover, following *StyleID* [5], we report a series of additional metrics in the new tables on **Page 16 (Tables 5–6)**.
* We have added a new section in the **Page 10 Section 4.3** to validate ***Stylos***'s controllable stylization capability through two interpolation experiments, and provided a new **Figure 6 on the Page 10**.
* We have fixed the font style issue by using the latest ICLR 2026 template.
* We have provided cleaner math formulas and algorithms of the different content-style coupling mechanisms, and images, scene, and 3D style losses.
* We have modified the quantitative and qualitative evaluation in **Section 4.2** so that it supports the newly added comparisons with additional stylization methods and evaluation metrics.
* We have included a section **A.1 Implementation** into the revised appendix page 14 for reproducibility.
* We have provided additional in-depth ablation analysis in **A.3 on Page 15**.
* We have added the 50 varying test style images into **Page 22-23**.

---

We thank all reviewers for their valuable suggestions, which motivated us to further clarify and strengthen the presentation of our work. Please read our individual response to each reviewer for details.

---

> ### Author Response · Authors · 2025-11-25
> **Reference**
>
> #### Reference
>
> [1] Kovács, Áron Samuel, Pedro Hermosilla, and Renata G. Raidou. "G‐Style: Stylized Gaussian Splatting." Computer Graphics Forum. Vol. 43. No. 7. 2024.
>
> [2] Zhang, Dingxi, et al. "Stylizedgs: Controllable stylization for 3d gaussian splatting." arXiv preprint arXiv:2404.05220 (2024).
>
> [3] Galerne, Bruno, et al. "SGSST: Scaling Gaussian Splatting Style Transfer." Proceedings of the Computer Vision and Pattern Recognition Conference. 2025.
>
> [4] Wang, Peng, Xiang Liu, and Peidong Liu. "Styl3R: Instant 3D Stylized Reconstruction for Arbitrary Scenes and Styles." arXiv preprint arXiv:2505.21060 (2025).
>
> [5] Chung, Jiwoo, Sangeek Hyun, and Jae-Pil Heo. "Style injection in diffusion: A training-free approach for adapting large-scale diffusion models for style transfer." Proceedings of the IEEE/CVF conference on computer vision and pattern recognition. 2024.
>
> [6] Wright, Matthias, and Björn Ommer. "Artfid: Quantitative evaluation of neural style transfer." DAGM German Conference on Pattern Recognition. Cham: Springer International Publishing, 2022.

---

### Meta-Review · Area_Chair_NeyN · 2026-01-06

**Summary:**

The paper presents Stylos, a method for 3D stylization conditioned on a reference style image, applicable to both unposed single-view and multi-view inputs. Stylos builds on the VGGT architecture, leverages 3D Gaussians as the scene representation, and performs style transfer in a single forward pass, achieving competitive visual quality with high efficiency. The initial reviews were very positive, and all reviewers agreed that the paper meets the acceptance bar. Reviewers praised the new capabilities enabled by Stylos and found the empirical results convincing. The minor concerns raised by reviewers were subsequently addressed in the rebuttal.

After carefully considering the paper, reviews, and rebuttal, the AC recommends acceptance.

**Reviewer Concerns:**

Most reviewers raised only minor concerns, primarily related to empirical validation and technical details. Reviewer Sv2n additionally questioned the technical novelty of Stylos in comparison to Styl3R. These concerns were largely addressed in the rebuttal, which clarified the technical details and presented additional experimental results. In particular, Reviewers Sv2n and LtqC acknowledged during the early discussion phase that their questions and concerns had been satisfactorily resolved.

**Reviewer Scores:**

The AC expects that all reviewers may maintain or raise their scores, given the reviews and the rebuttal.

---

### Decision · Program_Chairs · 2026-01-26

Accept (Poster)